# Impact of NLRP3 Depletion on Aging-Related Metaflammation, Cognitive Function, and Social Behavior in Mice

**DOI:** 10.3390/ijms242316580

**Published:** 2023-11-21

**Authors:** Elena D. Khilazheva, Angelina I. Mosiagina, Yulia A. Panina, Olga S. Belozor, Yulia K. Komleva

**Affiliations:** 1Research Institute of Molecular Medicine and Pathobiochemistry, Professor V. F. Voino-Yasenetsky Krasnoyarsk State Medical University, Krasnoyarsk 660022, Russiaangelina.mosiagina@gmail.com (A.I.M.);; 2Department of Biochemistry, Medical, Pharmaceutical and Toxicological Chemistry, Professor V. F. Voino-Yasenetsky Krasnoyarsk State Medical University, Krasnoyarsk 660022, Russia

**Keywords:** metaflammasome, immunosenescence, inflammasome, NLRP3, senescence-associated secretory phenotype, aging, inflammation

## Abstract

Immunosenescence and chronic inflammation associated with old age accompany brain aging and the loss of complex behaviors. Neuroinflammation in the hippocampus plays a pivotal role in the development of cognitive impairment and anxiety. However, the underlying mechanisms have not been fully explained. In this study, we aimed to investigate the disruption of insulin signaling and the mechanisms underlying metabolic inflammation (“metaflammation”) in the brains of wild-type (WT) and *NLRP3* knockout (KO) mice of different ages. We found a significant upregulation of the NLRP3 inflammasome in the hippocampus during aging, leading to an increase in the expression of phosphorylated metaflammation proteinases and inflammatory markers, along with an increase in the number of senescent cells. Additionally, metaflammation causes anxiety and impairs social preference behavior in aged mice. On the other hand, deletion of *NLRP3* improves some behavioral and biochemical characteristics associated with aging, such as signal memory, neuroinflammation, and metabolic inflammation, but not anxious behavior. These results are associated with reduced IL-18 signaling and the PKR/IKKβ/IRS1 pathway as well as the SASP phenotype. In *NLRP3* gene deletion conditions, PKR is down-regulated. Therefore, it is likely that slowing aging through various NLRP3 inhibition mechanisms will lessen the corresponding cognitive decline with aging. Thus, the genetic knockout of the NLRP3 inflammasome can be seen as a new therapeutic strategy for slowing down central nervous system (CNS) aging.

## 1. Introduction

In recent decades, the lifespan has increased dramatically, leading to a population growth of older people [1]. According to Khan et al., aging is a complex process that is influenced by genetic, epigenetic, and environmental factors and is characterized by changes at the molecular, cellular, and tissue levels [2]. Chronic inflammation, cell aging accompanied by the senescence-associated secretory phenotype (SASP), changes in glucose tolerance, and insulin resistance are some of the processes that determine aging [3]. Eventually, these pathophysiological processes lead to the development of age-related neurodegenerative disorders [4]. Immunosenescence and inflammation associated with old age accompany brain aging and the loss of complex behaviors, which is typical for Alzheimer’s disease (AD) and Parkinson’s disease (PD) [1,2]. A significant amount of recent research has concentrated on the links between the effects of overnutrition and obesity on brain aging and cognitive decline. Thus, it has been demonstrated that the molecular pathways underlying hippocampal functions, cognition, and memory are affected by insulin signaling [5]. We previously demonstrated that neuroinflammation caused by the overproduction of proinflammatory cytokines, activation of astroglia and microglia, and impaired reparative neurogenesis contributes significantly to the development of brain insulin resistance [4,6,7,8].

Insulin resistance, a hallmark of metabolic syndrome and metabolic dysfunction, has also been associated with neuroinflammation. Insulin signaling pathways are important for maintaining neuronal health and regulating inflammation in the central nervous system (CNS). Impaired insulin signaling can lead to dysregulation of immune responses and increased production of proinflammatory cytokines in the brain. Neuroinflammation, in turn, can further exacerbate metabolic dysfunction. Inflammatory molecules can interfere with insulin signaling in the brain, leading to insulin resistance and impaired glucose metabolism. This creates a vicious cycle where metabolic dysfunction promotes neuroinflammation, which in turn worsens metabolic dysfunction [9,10].

Various exogenous pathogen-associated molecular patterns (PAMPs) and damage-associated molecular patterns (DAMPs) can cause inflammation, which is an immunological response. Neutrophils, monocytes, and macrophages are part of the body’s innate immune system, which serves as the initial line of defense against infections [11]. Through pattern recognition receptors like Toll-like receptors (TLRs) and nucleotide-binding oligomerization domain-like receptors (NOD-like receptors or NLRs), these cells detect PAMPs and DAMPs and react accordingly. Reduced metabolic rates can dysregulate immune responses, leading to increased production and release of proinflammatory cytokines. These cytokines, such as interleukin-1β (IL-1β) and interleukin-18 (IL-18), can act as secondary signals to activate the inflammasome. PAMPs, derived from persistent infections or dysbiotic microbiota, can also contribute to sustained immune activation and inflammasome activation [12,13]. Inflammasomes, which are large cytosolic multimeric protein complexes and members of the NLR family, process and catalyze the maturation of proinflammatory cytokines like pro-IL-1 and pro-IL-18 [13]. The mature cytokines IL-1 and IL-18 alert the immune system to potential threats and boost the proinflammatory immune response. Numerous studies have concentrated on figuring out the precise mechanisms that cause NLR family pyrin domain-containing 3 (NLRP3) inflammation to be activated because the inflammatory stimuli that activate it can take many different forms. These studies have demonstrated that cellular metabolism plays a significant regulatory role in the NLRP3 inflammasome [14,15]. A growing body of research indicates that cellular metabolism plays a crucial role in cellular function, including the metabolic regulation of inflammation and macrophage polarization. The pathology of both acute and chronic conditions has been linked to inappropriate or excessive inflammatory activity, despite the NLRP3 inflammasome’s proposed protective role and inflammation in general. Chronic inflammation is typically thought of as low-intensity, long-term inflammation. This kind of inflammation, along with the metaflammation from metabolic sources, accelerates the aging process [15,16].

The link between metabolic dysfunctions and NLRP3 neuroinflammasome activation has been established in recent literature. Several studies have demonstrated that metabolic disorders, such as obesity, insulin resistance, and impaired glucose metabolism, can contribute to the activation of the NLRP3 inflammasome and subsequent neuroinflammation. Recent studies have shed light on the relationship between metabolic dysfunction and NLRP3 neuroinflammasome activation. One key aspect is the role of chronic low-grade inflammation associated with metabolic disorders, which contributes to NLRP3 inflammasome activation and subsequent neuroinflammatory responses.

A recent study investigated the impact of metabolic dysfunctions on NLRP3 inflammasome activation in the context of obesity. It was demonstrated that high-fat diet-induced obesity leads to NLRP3 inflammasome activation in adipose tissue and subsequently promotes systemic inflammation. The activation of the NLRP3 inflammasome was shown to be driven by metabolic dysfunction, including adipose tissue inflammation and insulin resistance [17].

In the context of neuroinflammation and neurodegenerative diseases, it was recently demonstrated that there is a connection between metabolic dysfunction and NLRP3 inflammasome activation. A number of recent studies confirmed the role of NLRP3 inflammasome activation in AD and identified metabolic dysfunction as a crucial factor. It was also highlighted that dysregulated glucose metabolism and impaired mitochondrial function contribute to NLRP3 inflammasome activation, neuroinflammation, and cognitive decline in AD. Furthermore, it was shown that metabolic dysfunctions, including obesity, insulin resistance, adipose tissue inflammation, lipid accumulation, and dysregulated glucose metabolism, can trigger NLRP3 inflammasome activation in various tissues, including the liver and the brain. This activation leads to the release of proinflammatory cytokines and the progression of neuroinflammatory processes [18].

Metabolic disorders are thought to be risk factors for brain aging. Therefore, impaired glucose metabolism or a reduction in the supply of glucose to the brain may occur along with the process of brain aging. Additionally, it was discovered that brain insulin resistance was linked to a higher risk of dementia and cognitive decline [19]. Not only do neurodegenerative diseases cause the multiprotein inflammasome to assemble, but metabolic disorders do as well [20,21]. Metaflammasomes or “metabolic inflammasomes” refer to the inflammation that metabolic disorders bring about. In other words, the cell’s metaflammasome is a DAMP-induced signaling cascade that results in a response from the metabolic pathway and the release of cytokines [22,23]. It has been established that the phosphorylated Ikappa-B kinase (IKK), insulin receptor substrate 1 (IRS1), c-Jun N-terminal kinase (JNK), and protein kinase R (PKR) are expressed in the human brain along with the other four main components of the metaflammasome complex [24].

Moreover, several studies have investigated the impact of metabolic rates on brain health and cognitive function. Epidemiological data have shown that individuals with lower metabolic rates are more prone to developing cognitive impairments and neurodegenerative diseases [25]. Thus, a decline in metabolic rate was associated with an increased risk of developing AD later in life. Similarly, impaired glucose metabolism and insulin resistance, indicative of lower metabolic rates, have been linked to a higher risk of cognitive decline and dementia [26,27].

Furthermore, clinical studies have demonstrated the association between metabolic dysfunctions, neuroinflammation, and cognitive decline. For instance, a study by Hsu et al. found that elevated markers of neuroinflammation, such as cytokines and chemokines, were associated with lower cognitive performance in individuals with metabolic syndrome [28].

The interaction between metabolic rates, neuroinflammation, and cognitive decline underscores the importance of maintaining optimal metabolic function for brain health. Therefore, approaches targeting metabolic pathways and inflammation could be promising for improving cognitive function and reducing neuroinflammation. Lower metabolic rates, often associated with metabolic dysfunctions, are linked to an increased risk of neuroinflammation and cognitive impairments. Understanding and addressing the interplay between metabolism, neuroinflammation, and cognitive function is essential for developing strategies to promote brain health and prevent age-related neurodegenerative diseases [11].

In light of this, it is imperative to conduct fundamental scientific research on the mechanisms that underlie neuroinflammation and metabolism to regulate and restore cognitive functions effectively. Furthermore, exploring these mechanisms can expand the brain’s regenerative potential and plasticity. Previous studies have shown that the activation of the NLRP3 inflammasome can regulate cell metabolism, underscoring the interconnection between inflammatory pathways and metabolic processes [29]. Therefore, the present study aimed to investigate the disruption of insulin signaling and the mechanisms of metabolic inflammation, also known as “metaflammation”, in the brains of wild-type (WT) and *NLRP3* knockout (KO) mice across different age groups.

## 2. Results

### 2.1. Aging Is Accompanied by an Increase in Phosphorylated Metaflammasome Proteinase Expression; Deletion of NLRP3 Does Not Lead to an Increase in the Expression of Metaflammasome Components

As previously mentioned, the metaflammasome is a DAMP-induced signaling cascade in the cell, followed by a metabolic pathway response and cytokine release [23]. Recently, it has been shown that double-stranded RNA-dependent protein kinase (PKR) and inhibitor of nuclear factor Ikappa-B kinase subunit beta (IKKβ) are central components of the NLRP3 metaflammasome complex and the main regulators of its activation [30,31]. To study the processes of metaflammation with aging and the role of NLRP3 in them, we investigated the expression of IKKβ and PKR in the brains of mice of different ages and genotypes. In this study, we used C57BL/6 mice, males aged 4–5 months (adult mice) and 14–15 months (aged mice), and *NLRP3* KO mice (B6.129S6-*NLRP3*^tm1Bhk^/J^J^) of the same ages.

It was found that the expression of phosphorylated IKKβ in vivo was significantly higher in aged WT mice (13164 ± 787.32 µm^2^) compared to adult mice of the same genotype (5851 ± 669.9 µm^2^) (*p* < 0.0003, Tukey’s test) (Figure 1A,B). Two-way ANOVA analysis revealed a statistically significant effect of the interaction between two factors (genotype and age) (F(1,21) = 26.35 *p* < 0.0001), as well as each factor separately: the effect of genotype F(1,21) = 51.25, *p* < 0.0001, the effect of age F(1,21) = 18.69, *p* = 0.0003. When comparing aged KO mice (4284.9 ± 869.96 µm^2^) and adult KO mice (3658.14 ± 686.42 µm^2^), it was found that aging in the absence of inflammasomes does not lead to a change in the expression of IKKβ kinase (*p* = 0.9481, Tukey’s test). When analyzing the expression of PKR protein kinase in vivo, a significant influence of genotype (F(1,16) = 26.65, *p* < 0.0001), age (F(1,16) = 7.067, *p* = 0.0172), and the interaction of these two factors was found (F(1,16) = 10.81, *p* = 0.0046). The area of PKR expression in the hippocampus of aged WT mice was significantly higher (8343 ± 1856.21 µm^2^) compared to adult animals (2566.26 ± 559.33 µm^2^) (*p* = 0.0034, Tukey’s test) (Figure 1C,D). Statistically significant differences in PKR expression were also found when comparing aged mice of different genotypes. In *NLRP3* KO mice, the area of PKR expression was significantly lower (133.67 ± 27.42 µm^2^) than in WT animals (8343 ± 1856.21 µm^2^) (*p* = 0.0001, Tukey’s test).

When analyzing the fluorescence intensity of phosphorylated IKKβ in cell cultures, it was also found that in the cells isolated from aged WT animals, the expression level of IKKβ kinase was significantly higher (12.3 ± 0.69 a.u.) than in cells obtained from adult animals (8.36 ± 0.51 a.u.) (*p* = 0.0006, Tukey’s test) (Figure 2A,B). Two-way ANOVA revealed statistically significant effects of genotype (F(1,16) = 35.78, *p* < 0.0001) and age (F(1,16) = 20.45, *p* = 0.0003). When comparing the level of phosphorylated IKKβ expression in the cells obtained from aged KO mice (7.56 ± 0.64 a.u.) and KO adult mice (6.5 ± 0.22 a.u.), no differences were found (*p* = 0.56, Tukey’s test), which correlates with the obtained in vivo data. Thus, aging was accompanied by an increase in the phosphorylated form of IKKβ in WT mice, but not in *NLRP3* KO mice. In addition, in the cells isolated from aged WT mice, the number of cells expressing PKR was five times higher (6.42 ± 0.26%) than in the cells isolated from adult WT animals (1.22 ± 0.21%) (*p* < 0.0001, Tukey’s test) (Figure 2C,D). In the cells isolated from *NLRP3* KO mice, an increase in PKR expression with age was also detected (1.2 ± 0.14% in the adult group, 3.7 ± 0.23 in the aged group, *p* < 0.0001, Tukey’s test). However, the number of cells expressing PKR in the experimental group of aged *NLRP3* KO mice was significantly lower than in the group of aged WT animals: 3.7 ± 0.23% for KO animals, 6.42 ± 0.26% for WT animals (*p* < 0.0001, Tukey’s test). In general, two-way ANOVA showed statistically significant effects of genotype (F(1,16) = 38.41, *p* < 0.0001), age (F(1,16) = 328.6, *p* < 0.0001), and the interaction of these two factors (F(1,16) = 37.27, *p* < 0.0001). Thus, aging was marked by an increase in the expression of phosphorylated metaflammasome proteases in WT animals but not in *NLRP3* KO mice.

### 2.2. Aging Is Accompanied by an Increase in Senescent Cells and Inflammatory Marker Expression; Deletion of NLRP3 Prevents Development of SASP Phenotype with Age

A limited ability to replicate is a defining characteristic of most normal cells and culminates in aging. It is known that senescent cells are not stimulated to divide by serum addition or passage in culture, and the aging process itself induces a specific cell cycle profile. Aging is characterized by increased cell size, expression of pH-dependent β-galactosidase activity, and altered patterns of gene expression [32,33,34]. Therefore, we studied the activity of β-galactosidase in hippocampal slices and cell cultures using the SA-β-Gal kit. Chromogenic cell staining confirmed the induction of SA-β-Gal in the hippocampus during aging in WT mice but not in *NLRP3* KO animals. Two-way ANOVA revealed a significant influence of the interaction between factors (age and genotype) (F(1,16) = 8.98, *p* = 0.0085). Multiple comparisons using the Tukey’s test confirmed that the cells in the hippocampus of aged WT mice showed a higher level of aging markers (3.56 ± 0.84% of the area of senescent cells) compared to the group of adult WT animals (1.02 ± 0.23% of the area of senescent cells) (*p* = 0.0172, Tukey’s test) (Figure 3). When comparing the groups of aged *NLRP3* KO mice and aged WT mice, it was shown that the area of senescent cells during aging in *NLRP3*^−/−^ mice was lower (0.83 ± 0.15% of the area of senescent cells) compared to aged WT animals (3.56 ± 0.84% of the area senescent cells) (*p* = 0.0164, Tukey’s test).

These results were also confirmed by the in vitro data obtained in the co-cultures of neurons and astrocytes from animals of different groups. Two-way ANOVA showed a significant effect of the interaction between factors (age and genotype) (F(1,20) = 11.3, *p* = 0.0031), as well as the effects of age (F(1,20) = 41.69, *p* < 0.0001) and genotype (F(1,20) = 31.39, *p* < 0.0001). Multiple comparisons confirmed that the cells isolated from aged mice exhibited higher activity of SA-β-Gal (optical density 0.083 ± 0.005 a.u.) compared to the group of adult animals (optical density 0.052 ± 0.001 a.u.) (*p* < 0.0001, Tukey’s test) (Figure 4). When comparing experimental groups of the cells obtained from aged mice of different genotypes, it was shown that β-galactosidase activity during aging in *NLRP3* KO mice was lower (optical density 0.055 ± 0.002 a.u.) compared to WT animals (optical density 0.082 ± 0.001 a.u.) (*p* < 0.0001, Tukey’s test). Thus, the activity of β-galactosidase in the cells of aged animals was higher, which is associated with an increase in senescent cells. At the same time, the obtained data indicate a direct role of the *NLRP3* gene in the formation of senescent cells.

In this study, we also confirmed the expression of the astrocyte marker GFAP and its colocalization with the inflammasome marker NLRP3. To confirm the accurate colocalization of these markers, we performed overlap index analysis using a confocal microscope. The Appendix A contains high-quality images and overlap index, providing visual and quantitative evidence of the colocalization between GFAP and NLRP3 (Appendix A).

The proinflammatory cytokines IL-1β, TNFα, and IL-18 are known to be the main constituents of SASP [4]. Therefore, we studied changes in the expression of IL-1β, as its maturation and secretion are primarily associated with the NLRP3 inflammasome. When analyzing the expression of IL-1β in hippocampal homogenates, we found a significant effect of the genotype (F(1,24) = 46.58 *p* < 0.0001). In multiple comparisons, it was found that in the hippocampal homogenates of adult WT animals, IL-1β was significantly higher (32.85 ± 6.51 pg/mg) compared to the group of adult *NLRP3* KO mice (4.65 ± 0.87) (*p* = 0.0004, Tukey’s test) (Figure 5). In addition, significant differences were found in the groups of aged mice of different genotypes: WT mice had a significantly higher IL-1β expression compared to KO animals (34.74 ± 6.26 pg/mg and 4.13 ± 1.46 pg/mg, respectively) (*p* = 0.0003, Tukey’s test). In the groups of both WT and *NLRP3* KO mice of different ages, we did not register any significant differences in the expression of IL-1β with ELISA.

When analyzing the expression of IL-18 in hippocampal slices, we found a significant effect of the interaction between age and genotype (F(1,16) = 328, *p* < 0.0001), as well as the effects of age (F(1,16) = 317.3, *p* < 0.0001) and genotype (F(1,16) = 478.4, *p* < 0.0001). In multiple comparisons, it was found that in the hippocampal cells of aged WT animals, the expression area of IL-18 was significantly higher (2038 ± 89.24 µm^2^) compared to the group of adult WT mice (283.1 ± 35.55 µm^2^) (*p* < 0.0001, Tukey’s test) (Figure 6). Also, significant differences were found in the groups of aged mice of different genotypes: WT mice had a significantly larger area of IL-18 expression (2038 ± 89.24 µm^2^) compared to KO animals (84.9 ± 14.9 µm^2^) (*p* < 0.0001, Tukey’s test). In the groups of *NLRP3* KO mice of different ages, significant differences in IL-18 expression were not registered, and the expression of NLRP3 was not detected by immunohistochemistry. Similar results were obtained when NLRP3 inflammasome expression images were analyzed in aging WT mice (1544 ± 273.5 μm^2^) versus in adult mice (672.3 ± 91.49 μm^2^) (*p* = 0.0079, Mann–Whitney criterion) (Figure 6). No NLRP3 expression was detected in *NLRP3* knockout groups of different age mice by IHC.

In our cell culture experiments, we also analyzed the expression of NLRP3 and IL-18. The results obtained in cell culture showed similar patterns of expression as observed in the in vivo data. Thus, when analyzing the expression of IL-18, two-way ANOVA showed a significant effect of the interaction between factors (age and genotype) (F(1,16) = 60.32, *p* < 0.0001), as well as the effects of age (F(1,16) = 64.53, *p* < 0.0001) and genotype (F(1,16) = 546.0, *p* < 0.0001). Multiple comparisons showed that in the cells obtained from aged WT animals, the intensity of IL-18 fluorescence was significantly higher (9.86 ± 0.41 a.u.) compared to the cells isolated from adult mice (5.12 ± 0.34 a.u.) (*p* < 0.0001, Tukey’s test) (Figure 7C,D). In addition to that, significant differences were also found in the experimental groups of cells obtained from aged mice of different genotypes: IL-18 fluorescence intensity was significantly higher in the group of WT mice (9.86 ± 0.41 a.u.) compared to the group of KO animals (0.44 ± 0.19 a.u.) (*p* < 0.0001, Tukey’s test). Furthermore, the data on NLRP3 expression changes in vivo were also confirmed in vitro. In the cell cultures isolated from *NLRP3* KO mice of different ages, NLRP3 expression was not detected. In the cells obtained from aged WT mice, the intensity of NLRP3 fluorescence was significantly higher (15.64 ± 1.06 a.u.) compared to the cells isolated from adult animals (5.92 ± 0.32 a.u.) (*p* < 0.0001, Tukey’s test) (Figure 7A,B). Thus, with aging, there is an increase in senescent cells with a secretory phenotype accompanied by an increase in the expression of NLRP3 inflammasomes and IL-18.

We then evaluated HMGB1 expression in cell culture. It is known that HMGB1 is a key protein associated with DNA stability, gene expression, and inflammation and is considered one of the components contributing to the pro-inflammatory milieu typical of the SASP. HMGB1 is presumed to hold a pivotal role in the initial phase of IL-1β and IL-18 release, potentially priming the subsequent cascade of cytokine activities, leading to an exaggerated release of IL-1β and IL-18. This hyper-release, in turn, may instigate inflammatory responses and contribute to the advancement of neurodegeneration [4].

Analyzing the HMGB1 expression, two-way ANOVA showed a significant effect of age (F(1,20) = 89.61, *p* < 0.0001) and genotype (F(1,20) = 5.05, *p* = 0.03). Multiple comparisons showed that in the cells obtained from aged WT animals, the intensity of HMGB1 fluorescence was significantly higher (6.32 ± 0.49 a.u.) compared to the cells isolated from adult mice (1.92 ± 0.31 a.u.) (*p* < 0.0001, Tukey’s test) (Figure 8A,B). In addition to that, significant differences were also found in the experimental groups of cells obtained from aged mice of different genotypes: HMGB fluorescence intensity was significantly higher in the group of WT mice (6.32 ± 0.46 a.u.) compared to the group of KO animals (4.43 ± 0.26 a.u.) (*p* = 0.0045, Tukey’s test). Meanwhile, protein expression also increases with age in knockout mice: adult *NLRP3* KO (2.23 ± 0.29 a.u.) vs. aged *NLRP3* KO (4.43 ± 0.26 a.u.) (*p* = 0.0013 Tukey’s test).

### 2.3. Deletion of NLRP3 Leads to an Increase in Lactate Levels but Not to a Change in the Expression of Insulin and Insulin Resistance Markers

Since it is known that the process of brain aging can be accompanied by impaired glucose metabolism or reduced glucose supply to the brain, central insulin resistance is associated with an increased risk of cognitive deficit [5]. Hence, we studied the expression of insulin receptors (IR) in the co-culture of astrocytes and neurons obtained from experimental animals. Although there was a trend towards an increase in the expression of IR in cells isolated from aged animals (9.94 ± 0.91 a.u.) compared to adult mice (8.56 ± 0.23 a.u.), we did not detect any statistically significant differences (*p* = 0.26, Tukey’s test) (Figure 9).

Next, we examined the concentration of a glucose metabolism product, lactate, in the hippocampus of experimental animals using an enzyme immunoassay. Two-way ANOVA revealed a significant influence of genotype (F(1,21) = 16.96, *p* = 0.0005) in the studied groups. Thus, we registered a high level of lactate in adult *NLRP3* KO mice (22.2 ± 3.2 nmol/µg) compared to age-matched WT animals (5.95 ± 0.69 nmol/µg) (*p* = 0.0011, Tukey’s test) (Figure 10A). No significant differences in insulin expression were observed during the assessment of insulin concentration in the hippocampus among the WT groups and *NLRP3* knockout mice of different ages. The statistical analysis, including the interaction between genotype and age (F(1,20) = 1.381, *p* = 0.2538, two-factor ANOVA), as well as the genotype factor (F(1,20) = 0.3136, *p* = 0.5817) and age factor (F(1,20) = 3.788, *p* = 0.0658), did not reveal any significant influence on the insulin levels (Figure 10B).

Immunohistochemical study of the expression area of pathologically phosphorylated IRS1 did not reveal statistically significant differences in the studied groups ((F(1,24) = 1.165, *p* = 0.2912—age; F(1,24) = 0.02554, *p* = 0.8744—genotype; F(1,24) = 1.615, *p* = 0.2160—interaction, two-way ANOVA): in the groups of WT animals, 3149 ± 277.02 µm^2^ for aged and 3086.7 ± 318.54 µm^2^ for adult mice (*p* = 0.9991, Tukey’s test); in the groups of *NLRP3* KO animals, 3781.83 ± 488.86 µm^2^ for adult and 2519.3 ± 579.75 µm^2^ for aged mice (*p* = 0.3650, Tukey’s test) (Figure 11). Also, there were no statistically significant differences in the fluorescence intensity of phosphorylated IRS1 in the cell cultures obtained from animals of different experimental groups: in the groups of WT animals, 10.72 ± 1.13 a.u. for aged and 13.36 ± 1.02 a.u. for adult mice (*p* = 0.28, Tukey’s test); in the groups of *NLRP3* KO animals, 10.84 ± 0.98 a.u. for adult and 11.68 ± 0.85 a.u. for aged mice (*p* = 0.93, Tukey’s test) (Figure 12). Thus, neither with aging nor with *NLRP3* KO were the signs of insulin resistance revealed. However, in the hippocampus of *NLRP3* KO mice, an increase in lactate levels was registered.

### 2.4. Aging and NLRP3 Deletion Cause Alterations in Synaptic Transmission

Lactate, which enters activated synapses, plays a metabolic role in meeting the high energy needs of membrane and cellular processes necessary for long-term synaptic plasticity and memory [35,36]. Therefore, we studied whether increased levels of lactate in NLRP^−/−^ mice would lead to any changes in synaptic plasticity. An electrophysiological study revealed an increase in the amplitude of field excitatory postsynaptic potentials (fEPSP) in aged WT mice (0.49 ± 0.06 mV) compared to adult WT mice (0.32 ± 0.022 mV) (*p* = 0.0433, Tukey’s test) (Figure 13A), whereas the rise time in aged mice of the same genotype was significantly shorter compared to adult mice (1.18 ± 0.17 ms and 2.72 ± 0.33 ms, respectively) (*p* = 0.0009, Tukey’s test) (Figure 13B). An increased fEPSP amplitude was also observed in adult *NLRP3*^−/−^ mice (0.49 ± 0.069 mV) compared to adult WT mice (*p* = 0.0364, Tukey’s test) (Figure 13A). Furthermore, the rise time in adult *NLRP3*^−/−^ mice was 1.2 ± 0.15 ms faster compared to adult WT mice (2.73 ± 0.33 ms) (*p* = 0.0008, Tukey’s test) (Figure 13B). In aged WT and KO mice, no differences were observed in the fEPSP amplitude and rise time. Accordingly, the observed changes in fEPSP amplitude and rise time in aged WT and adult *NLRP3* KO mice indicate altered synaptic transmission in these experimental groups, which is probably associated with alterations in neurotransmitter release from a presynapse.

### 2.5. Aging in Mice Is Accompanied by the Development of Anxiety-like Behavior and Impaired Social Preference Activity Whereas NLRP3 Deletion Leads to Anxious Behavior in Adult Animals

The observed molecular and electrophysiological changes in mice with *NLRP3* KO compared to WT animals raised the question of what behavior would be observed in animals with this genotype. First, we investigated the change in locomotor activity caused by aging and *NLRP3* KO by using the open field test performed on adult KO mice in comparison with aged mice. No significant differences were found in the total distance walked in the open field in WT and *NLRP3*^−/−^ groups, indicating that the overall locomotor activity was the same in all experimental groups (Figure 14). However, *NLRP3* deletion in adult mice led to an increase in the time spent in the central zone of the arena (42.95 ± 6.845 s) compared to WT mice of the same age (17.05 ± 4.157 s) (*p* = 0.0120, Tukey’s test). Two-way ANOVA showed a significant effect of genotype (F(1,18) = 11.35, *p* = 0.0034). Nevertheless, the time spent around the perimeter of the open field did not differ between all experimental groups.

Next, we studied the fear of novelty in WT mice compared to *NLRP3* KO animals. For this, a new non-social (non-living) object was placed in the center of an open field arena to evaluate the following parameters: the time spent with the object, the time spent in the center, and the number of entrances into the area with the new object (Figure 15). Interestingly, aged WT mice showed significantly less activity in the open field (26.778 ± 2.068 m) compared to KO mice of the same age (40.267 ± 2.336 m) (*p* = 0.0079, Tukey’s test) and compared to adult WT mice (36.531 ± 2.091 m) (*p* = 0.0079, Tukey’s test). Two-way ANOVA revealed a significant effect of genotype (F(1,17) = 6.342, *p* = 0.0221) as well as the interaction between factors (genotype and age) (F(1,17) = 9.987, *p* = 0.0057). Also, aged WT mice spent more time in the periphery (383.620 ± 66.169 s) than adult mice of the same genotype (237.317 ± 9.270 s) (*p* = 0.0352, Tukey’s test). A significant effect of the interaction between factors (genotype and age) (F(1,17) = 4.661, *p* = 0.0454) was found with a two-way ANOVA test. The time in the medium zone of the arena was also longer in aged mice with *NLRP3* deletion (290.725 ± 10.802 s) compared to age-matched WT mice (160.060 ± 46.458 s) (*p* = 0.0207, Tukey’s test), and there was a significant effect of genotype (F(1,17) = 9.196, *p* = 0.0075). When analyzing the time spent in the area with the object, significant influences of genotype (F(1,17) = 8.237, *p* = 0.0106) and age (F(1,17) = 6.58, *p* = 0.0201) were revealed. Adult WT mice spent significantly more time with the new object (123.333 ± 18.123 s) when compared to KO animals of the same age (43.150 ± 13.966 s) (*p* = 0.0111, Tukey’s test) and aged WT mice (48.380 ± 20.858 s) (*p* = 0.0248, Tukey’s test). Thus, adult WT mice were more active and showed interest in the non-social object. With age, this activity decreased. However, there was no such pattern in *NLRP3* KO mice as we did not observe such differences between the animals of adult and aged groups. The observed phenotype may indicate the increased fear of novelty in *NLRP3*^−/−^ mice with preserved high motor activity.

To analyze if the same behavior would be observed towards a living object, we placed a social object (another individual) in the center of the arena (Figure 16). A similar trend was found with the social object as in the previous test. In aged WT mice, the total distance walked in the open field was reduced (17.29 ± 0.893 m) compared to KO animals (31.029 ± 2.145 m) (*p* = 0.0004, Tukey’s test). Significant influences of genotype (F(1,17) = 31.96, *p* < 0.0001) and the interaction between factors (genotype and age) (F(1,17) = 5.512, *p* = 0.0313) were revealed. Also, the time spent at the periphery increased with age in WT mice (403.067 ± 46.922 s) compared to adult WT animals (208.733 ± 26.031 s) (*p* = 0.0059, Tukey’s test). This may indicate the development of anxiety and fear of the social object. The influence of age was significant (F(1,18) = 15.39, *p* = 0.0010). The time spent with the social object also differed: the effect of factors interaction (genotype and age) (F(1,18) = 5.868, *p* = 0.0262), age—(F(1,18) = 27.04, *p* < 0.0001), genotype—(F(1,18) = 9.885, *p* = 0.0056). During aging in WT mice, the time with another individual significantly decreased (40.883 ± 10.143 s) compared to adult mice (161.633 ± 21.765 s) (*p* = 0.0001, Tukey’s test). Furthermore, adult *NLRP3* KO mice also spent less time with the living object (73.450 ± 13.766 s) than WT mice of the same age (161.633 ± 21.765 s) (*p* = 0.0029, Tukey’s test). Thus, it can be assumed that aging in WT mice was accompanied by the development of anxiety and impaired social activity. However, no such changes were found in aged mice with a deletion of the *NLRP3*^−/−^ gene as even adult *NLRP3* KO animals showed anxious behavior and this phenomenon did not change with age.

### 2.6. NLRP3 Knockout Mice Spent Less Time with Social Object Than Age-Matched Wild Type Mice; Aging in Absence of NLRP3 Inflammasome has no Influence on Sociability but Decreases Social Preference for Novelty

Mice were tested in the three-chamber apparatus for social interactions. Sociability is defined as the experimental mouse spending more time in the chamber containing the novel mouse than in the chamber containing the novel object. Analysis revealed that adult WT mice spent significantly more time in the chamber with social objects (77.86 ± 3.3 (%)) than *NLRP3* KO mice (60.55 ± 1.4 (%)) (*p* = 0.0152, Tukey’s test). There was no difference between different ages in the *NLRP3* KO group—adult (60.55 ± 1.4 (%)) vs. aged (48.33 ± 5.9 (%)) (*p* = 0.1930, Tukey’s test). Significant influences of age (F(1,22) = 13.75, *p* = 0.0012) and genotype (F(1,22) = 14.88, *p* = 0.0009) were revealed in the test of sociability. However, adult WT mice spent more time with a new social stranger (81.22 ± 5.6 (%)) than aged WT mice (33.03 ± 3.24 (%)) (*p* < 0.0001, Tukey’s test) and adult *NLRP3* KO mice (61.99 ± 3.12 (%)) (*p* = 0.0179, Tukey’s test). Significant influences of age (F(1,20) = 90.46, *p* < 0.0001) and genotype (F(1,20) = 6.332, *p* = 0.0205) as well as the interaction between age and genotype (F(1,20) = 4.533, *p* = 0.0459) were revealed. Aging in *NLRP3* KO mice also leads to a decrease in social preference: % preference for stranger 2 in *NLRP3* KO aged (31.43 ± 4.12 (%)) was lower than in *NLRP3* KO adult mice (61.99 ± 3.12 (%)) (*p* = 0.0002, Tukey’s test). Data on sociability and social preference are presented for both groups in Figure 17.

### 2.7. Aging Does Not Affect Signaling Memory in NLRP3 KO Mice

Next, we studied the memory of fear in the fear conditioning test in mice. On the first day of testing (conditioning day), there was no significant effect of the interaction of factors (F(9,87) = 0.7239, *p* = 0.6857) and the group factor (F(3,29) = 1.909, *p* = 0.1501) when comparing adult and aged mice. However, the significant effects of the time factor (stimuli CS-US1- CS-US3 (F(3,87) = 112.9, *p* < 0.0001)) and object matching (F(29,87) = 3.844, *p* < 0.0001) within all ages groups in WT and *NLRP3* KO mice were revealed by two-way ANOVA (Figure 18A).

On the second day of testing (contextual), we compared the freezing time between four groups: adult WT, aged WT, adult *NLRP3*^−/−^, and aged *NLRP3*^−/−^ mice. Two-way ANOVA showed a statistically significant influence of age (F(1,26) = 8.17, *p* = 0.0083) but not genotype (F(1,26) = 0.04482, *p* = 0.8340) or the interaction between these two factors (F(1,26) = 0.8936, *p* = 0.3532). Statistically significant differences were registered between adult (86.447 ± 2.03%) and aged (65.206 ± 5.153%) WT mice (*p* = 0.0492, Tukey’s test). It was found that the freezing time was reduced in aged WT animals, which was caused by a decrease in the perception of the environment (context) as a potentially dangerous place and, as a result, a violation of memory processes in a particular context. In *NLRP3*^−/−^ mice, the freezing time did not change at different ages, which probably indicates the absence of an effect of age on contextual memory in *NLRP3* KO animals (Figure 18).

On the third day of testing (cued) in a new setting, significant effects of the group factor (F(3,30) = 4.662, *p* = 0.0086), the tone factor (F(3,30) = 92.85, *p* < 0.0001), and object matching (F(30,30) = 3.634, *p* = 0.0003) were revealed. Similar to the observations on the second day, a reduction in the freezing time was recorded in aged WT animals (69.16 ± 6.05%) compared to adult WT mice (98.74 ± 0.94%) due to the absence of conditioned fear formation after the application of white noise (tone) in the new setting (*p* = 0.0106, Tukey’s test). The freezing time between adult and aged *NLRP3*^−/−^ mice, as well as on the second contextual day, did not differ, suggesting that age did not affect signaling memory in *NLRP3* KO mice (Figure 18C). Thus, it can be concluded that the *NLRP3* gene has a protective genotype during aging.

## 3. Discussion

The proportion of elderly people is rising along with life expectancy in contemporary society. Chronic inflammation is linked to aging and contributes to or exacerbates a number of age-related non-communicable diseases. Although the NLRP3 inflammasome is frequently known to promote pathological inflammation, the precise mechanisms that cause NLRP3 activation in aging and associated comorbidities are not well understood. [37]. In this study, we analyzed the behavioral, electrophysiological, molecular, and biochemical changes that occur with aging and the mechanisms by which they activate the NLRP3 inflammasome and the *NLRP3* gene’s function in the aging process.

First, we demonstrated that aging in WT mice is accompanied by an increase in the expression of the protein kinases IKK and PKR that have been phosphorylated. In aging animals, PKR was significantly upregulated in the hippocampus, which indicates that the hippocampal PKR/NLRP3 inflammatory pathway is crucial for the emergence of the senescent cell phenotype. Moreover, earlier studies demonstrated the protective role of *PKR* gene knockout. Thus, after induction of obesity with a high-fat diet in WT and *PKR*^−/−^ mice, a significant rise in brain metaflammasome protein activation was observed in WT mice but not in *PKR*^−/−^ animals [30]. Also, it was previously confirmed that through the production of reactive oxygen species (ROS) and MAP kinases ERK1/2, JNK, and p38, PKR causes the activation of the canonical inflammasome NLRP3 and caspase-1. It leads to the processing and secretion of pro-IL-1β [38]. Since PKR regulates the NLRP3 inflammasome along with the fact that *PKR* deletion reduces the levels of NLRP3, high mobility group 1 protein (HMGB1), and IL-1β in macrophages [39], the decrease in PKR expression in both adult and aged KO mice was presumable. Stress signals activate PKR, and this autophosphorylation causes the NF-κB pathway to be activated and the formation of inflammasomes. In *NLRP3* KO mice, this pathway is inactive, which means that NF-κB is not activated and PKR cannot be overexpressed. Also, PKR overexpression during aging in WT mice might be a mechanism explaining the impaired contextual memory in experimental animals. For instance, earlier research in mice and monkeys demonstrated that TNFα (tumor necrosis factor alpha) can induce PKR, which results in memory impairment [40]. PKR can activate a number of downstream pathways that result in an inflammatory, apoptotic, or autophagic response once it has been phosphorylated. Research on PKR knockout mice has demonstrated that PKR loss dramatically reduces inflammation [41]. Taken together, our data show that PKR is involved in the activation of NLRP3 during aging, and the deletion of *NLRP3* is accompanied by the inhibition of PKR, one of the components of the metaflammasome.

IKKβ was another member of the metaflammasome, and its expression was elevated in the hippocampus of aging WT mice. IκB proteins are phosphorylated by an activated IKK complex, which causes their subsequent degradation. The NF-κB dimer that has been trapped in the cytoplasm is consequently released and moved to the nucleus to activate a particular transcriptional mechanism. Numerous physiological processes depend on normal NF-κB activation, whereas chronic or abnormal activation is linked to inflammatory and age-related diseases. The expression of SASP factors, such as proinflammatory cytokines and chemokines (p16INK4a, p21CIP1, IL-6, IL-1 α, HMGB1, and TNF α), is mediated by NF-κB, which thus plays a significant role in cell senescence and aging. Senescence is also induced in mammals by constitutive activation of NF-κB [42,43]. Increased production of SASP factors may further enhance cellular senescence via cell-autonomous and/or non-autonomous mechanisms. Furthermore, it was shown that IKKβ is required for the rapid formation of the NLRP3 inflammasome and subsequent activation of caspase-1, gasdermin D, and IL-18 secretion [31]. In this study, we showed that constitutive activation of IKKβ is associated with aging and excessive activation of the NLRP3 inflammasome. In *NLRP3* KO mice, IKKβ expression was reduced and did not change in aged animals.

We also demonstrated both in vivo and in vitro that aging is accompanied by an increase in the number of senescent cells. Such cells are characterized by irreversible growth arrest in response to various stress factors and by the aging-associated secretory phenotype (SASP) [44]. A variety of stress conditions contribute to the progression of immunosenescence including chronic inflammation, which is considered to be a hallmark of aging and induces cellular senescence by increasing various proinflammatory cytokines, metabolites, aggregates, and reactive substances [4,43]. All of these molecules have been shown to trigger the NLRP3 inflammasome through various mechanisms and enhance the inflammatory response, which contributes to the further development of inflammation and disease progression [4,29]. One of the interesting findings of this study was that in adult *NLRP3* KO mice, the number of senescent cells was similar to that in WT animals. At the same time, in KO mice, the number of senescent cells did not increase but even decreased with age. Previous studies have shown the role of the NLRP3 inflammasome in several aging-related events. For instance, it has been demonstrated that *NLRP3* genetic deletion in mice lengthens lifespan by reducing a number of age-related degenerative changes, including impaired glycemic control, bone loss, cognitive dysfunction, and impaired motor activity [45]. In addition, the age-related increase in the number of myopathic fibers was prevented by the deletion of *NLRP3* in aged mice, which improved muscle strength and endurance [46]. Finally, aged WT animals showed elevated levels of IL-1β and active caspase-1 protein compared to *NLRP3*^−/−^ mice, and aged WT and *NLRP3*^−/−^ mice showed elevated levels of TNFα, IL-6, and IL-8 [47]. These data indicate that the loss of *NLRP3* does not affect the age-related enhancement of other inflammatory pathways. Inflammation itself certainly accompanies aging, so the formation of senescent cells in *NLRP3* KO mice can be explained by the activity of other inflammatory pathways which can lead to the overexpression of TNFα, IL-6, and IL-8 [47].

Another finding of the study was that *NLRP3* deletion leads to an increase in lactate levels. According to the literature, lactate exhibits a dual role, both fostering and hindering long-term potentiation (LTP), contingent on the specific brain region and neuron type. For example, in the CA3 area, lactate induces glutamatergic potentiation in recurrent collateral synapses, but not in the synapses of mossy fibers of CA3 pyramidal cells [35]. In CA1 neurons, lactate allows the induction of LTP [48]. However, in this study, the types of plasticity and long-term potentiation (LTP) or depression (LTD) were not considered, only the parameters of synaptic transmission.

It was previously reported that *NLRP3* KO mice have higher glucose levels compared to WT animals on a standard diet, and *NLRP3* KO mice have higher lactate levels regardless of the diet. This may indicate that the lack of NLRP3 may increase lactate dehydrogenase or glycolysis activity, which mediates the conversion of lactate to pyruvate and back, or a combination of these two mechanisms [49]. Recent evidence regarding obesity and diabetes suggests that metabolic inflammasomes, or metaflammasomes, mediate chronic inflammation [50,51]. Moreover, chronic inflammation is linked not only to metabolic diseases, such as obesity and type 2 diabetes, and Alzheimer’s disease but also possibly plays a role in physiological aging [30]. However, according to our data, aging in mice did not lead to severe metabolic disturbances. Thus, there was no significant change in the expression of insulin receptors, which suggests that during physiological aging, metabolic inflammation is observed earlier than insulin signaling impairment. Therefore, according to our observations, metabolic inflammation precedes insulin signaling impairment. The absence of significant changes in insulin receptor expression might suggest that during the aging process in mice, metabolic inflammation manifests as an early event before substantial alterations in insulin signaling pathways [52]. This could be due to the intricate interplay between inflammatory responses and metabolic dysregulation.

Aging is often associated with a low-grade, chronic inflammatory state, which can impact cellular processes. Metabolic inflammation may induce alterations in various molecular pathways, affecting cellular responses and potentially contributing to changes in insulin sensitivity before significant modifications in insulin receptor expression become apparent [4]. The presence of inflammatory mediators associated with aging might have a more immediate impact on cellular functions related to metabolism. These mediators, such as cytokines or inflammatory signaling pathways, could influence cellular responses, disrupting metabolic homeostasis earlier than impacting insulin receptor expression levels [53].

It is possible that cells might exhibit adaptive mechanisms to maintain insulin receptor expression levels even in the presence of metabolic inflammation [54]. However, these compensatory mechanisms might not fully mitigate the underlying functional changes in insulin signaling pathways, explaining the discrepancy between insulin receptor expression and the onset of metabolic disturbances. This potential explanation highlights the complex interplay between metabolic inflammation and insulin signaling during aging, suggesting a temporal divergence in the manifestation of these processes. Further investigation into the intricate molecular mechanisms underlying these observations could provide deeper insights into the aging process and its impact on metabolic and insulin signaling pathways. This nuanced understanding could be further explored through detailed molecular studies to elucidate the specific pathways and interactions contributing to these observations.

We also demonstrated electrophysiological changes in the brains of aged WT and *NLRP3* KO mice. Synaptic transmission involves the release of neurotransmitters from presynaptic neurons, which then bind to specific postsynaptic receptors. The amplitude and rise time of fEPSP indicate the strength of excitatory synaptic transmission: how quickly the neurotransmitter is released by the presynapse and binds to the receptor on the postsynapse [55]. The number of AMPA glutamate receptors in synapses is the main factor determining synapse strength and varies from synapse to synapse [56]. The strength of the fEPSP response also depends on the activation of other glutamate or GABAergic receptors. In addition, glutamate can be released not only by neurons but also by astrocytes and microglial cells, which is observed during inflammation [57,58]. Aging is known to impair microglial function with increased susceptibility to proinflammatory activation, thereby contributing to the development of aging-associated neurodegeneration [59,60]. The discussion centered on plausible factors influencing alterations in fEPSP parameters, encompassing variations in the number of AMPA receptors in the postsynapse, elevated neurotransmitter release by neurons or glial cells amidst an inflammatory environment, and potential impairment in neurotransmitter reuptake by glial cells. Observing a reduction in rise time and an augmentation in amplitude, it can be inferred that neurotransmitter release from the presynapse accelerates and reaches the receptors at an expedited rate. Moreover, evidence supports the notion that lactate may amplify currents flowing through AMPA receptors, thereby instigating the initiation of long-term potentiation [36]. In this case, a prolonged increase in lactate in adult *NLRP3* KO mice may lead to an increase in their fEPSP amplitude.

The fEPSP parameters can also be influenced by postsynaptic factors—the number of AMPA receptors or impaired reuptake of glutamate from the synaptic cleft by astrocytes and, as a consequence, stronger activation of postsynaptic receptors. An increase in the amplitude of fEPSP in aged animals may indicate the activation of inflammation, as a result of which the number of glutamate receptors on the postsynapse changes or glutamate is released by glial cells. A decrease in the rise time indicates a faster release of glutamate into the synaptic cleft or a faster interaction with receptors, which can also be associated with a change in the distribution of receptors on the postsynapse. NLRP3 inflammasomes have a dual role: physiological levels are necessary for the normal functioning of neurons, and excess activity leads to neurodegeneration. The complete shutdown of inflammasomes also negatively affects neurons as various mediators not only contribute to the development of inflammation but also change synaptic transmission and plasticity. For example, IL-1β is known to regulate long-term synaptic plasticity and may reduce glutamate reuptake by astrocytes [58,61]. The observed changes in adult and aged KO mice compared to adult WT mice may be due to the shutdown of intracellular cascades that may affect receptor activity or glutamate release. The obtained results indicate an increase in the excitability of hippocampal neurons and a change in neurotransmitter release from presynaptic terminals in synapses in aged and KO mice compared to adult WT animals.

The changes in complex behavior were also reflected in changes in the expression of NLRP3 inflammasomes in the brain. We have previously shown that *NLRP3* deletion has an impact on the emotional sphere and memory: the development of anxious behavior, which is reflected at the cellular and molecular levels as the disruption of neurogenesis, astrocyte formation, and synaptic transmission [7,8,62]. In studies on animals, stress causes an increase in IL-1β levels throughout the brain. It has been demonstrated by other research teams that IL-1β is necessary for healthy learning and memory because both low and high levels of IL-1β impair memory formation [63]. However, despite a growing body of evidence, it is still unknown whether the NLRP3 inflammasome is a causal factor and how it affects the memory of fear in aging, despite links between IL-1β production, neuroinflammation, the memory of fear, and related disorders. The NLRP3 inflammasome plays a significant role in the regulation of fear memory, including as people age, as we have demonstrated in this study. In this study, we examined aging-related anxiety, social behavior, memory formation, and memory retrieval in WT and *NLRP3* KO mice. It is known that episodic memory decline and changes in memory-related brain processes are linked to aging [64]. This study showed that aging mice had impaired contextual memory, but not acquisition or signaling memory. The amygdala of the brain plays a significant role in the development and expression of conditioned freezing, and the hippocampus is necessary for contextual memory, according to previous research [65]. Since the hippocampus plays an important role in regulating contextual fear, we particularly focused on this area of the brain in our experiments. According to earlier research, aging impairs one’s capacity to use context to generate learned responses to threats, possibly as a result of changes to brain regions that control behavior that depends on context and are particularly vulnerable as people age [66]. These structures primarily include the dentate gyrus of the hippocampus [67]. Thus, in aged mice, a violation of the contextual memory associated with the hippocampus was recorded. Also, aging in mice was accompanied by the development of anxiety and impaired social preference and activity. In mice, *NLRP3* deletion indicates alterations in emotional responses. *NLRP3* knockout mice exhibit changes in anxiety-like behaviors, as we have already described earlier [8]. These mice might spend more time in the periphery and less in central areas, which could indicate increased anxiety-like responses. These mice also display changes in social preference, such as altered social interaction patterns with novel strangers, that also could be associated with anxiety-like behavior. Therefore, understanding the impact of *NLRP3* deletion on social behaviors is an ongoing area of research. However, the freezing time between adult *NLRP3*^−/−^ and aged *NLRP3*^−/−^ mice, as well as on the second contextual day, did not differ, which may indicate the absence of an effect of age on signaling memory in *NLRP3* KO mice. Thus, *NLRP3*^−/−^ has a protective genotype during aging in terms of signaling memory and social patterns, but not in anxiety-like behavior. In contrast to the results obtained by Dong et al., we confirmed increased fear in *NLRP3*^−/−^ mice with preserved high motor activity [68]. These observations suggest potential implications of NLRP3 in cognitive and social functions.

Modulating the NLRP3 inflammasome activation might offer therapeutic avenues. Targeting this pathway could potentially mitigate age-related inflammation and associated diseases. For instance, developing small molecule inhibitors or biologics specifically aimed at NLRP3 activation might have relevance in treating conditions linked to age-induced inflammation.

Recent advances in understanding the NLRP3 inflammasome’s structure, inhibitor binding, and signaling have revealed its potential as a promising drug target for treating various disease conditions. Pharmacological targeting of NLRP3 has shown efficacy in ameliorating diseases in animal models, highlighting its importance in managing chronic health concerns globally [69].

Studies have demonstrated that inhibiting the NLRP3 inflammasome in mice results in protective effects against age-related issues, such as increased insulin sensitivity, reduced levels of certain markers linked to aging, and prevention of cardiac damage and abnormalities associated with aging. Inhibition of NLRP3 also showed promise in improving heart health, preserving function, and extending lifespan in aged mice [47]. Furthermore, research has indicated the involvement of the NLRP3 inflammasome in age-related female fertility decline, suggesting its potential as a therapeutic target to address infertility issues associated with aging [70].

It should be noted that the blocking of inflammasomes in experiments often shows a positive protective effect in many types of inflammation and age-associated diseases and conditions in models and cell cultures. However, the systemic effect of complete inhibition of the NLRP3 inflammasome may also have negative consequences. Thus, some of our previous and current data suggest a potential dual role of NLRP3, where its inhibition might lead to anxious behavior [7,8]. Future studies need to explore the nuances and optimal timing of NLRP3 modulation, potentially focusing on regulating excessive interleukin production and the development of the senescence-associated secretory phenotype. The reduction of the inflammatory response is considered a viable strategy for anti-aging interventions. It should be taken into consideration that cellular senescence plays a complex role, capable of offering compensatory benefits or hastening aging by limiting tissue regeneration. Considering these dual roles in the development of interventions becomes crucial, allowing a comprehensive understanding of their potential effects on the aging process and related diseases [71].

It is worth considering that medicine is undergoing a gradual conceptual shift from “treating disease” to “protecting health”. In other words, medicine is shifting from treatment after disease onset to intervention before recognized risk factors lead to disease onset. Defining the hallmarks of aging provides a framework for the mechanisms of aging and thereby theorizes the need to improve human health and longevity. With the ever-increasing trend of population aging, anti-aging strategy is undoubtedly an increasingly important area of focus for the reduction of inflammation by different approaches including target molecules, calorie restriction, and physical activity [4,71].

## 4. Materials and Methods

### 4.1. Animals

This study used C57BL/6 mice, males aged 4–5 months, weighing 25–30 g (*n* = 12); C57BL/6 mice, males aged 14–15 months, weighing 25–30 g (*n* = 15); *NLRP3* KO mice (B6.129S6-*NLRP3*^tm1Bhk^/J^J^) aged 4–5 months, weighing 25–30 g (*n* = 12); *NLRP3* KO mice (B6.129S6-*NLRP3*^tm1Bhk^/J^J^) aged 14–15 months, weighing 25–30 g (*n* = 15). We used only male mice to reduce the potential confounding effects of hormonal variations. An age-matched male B6.129S6-Nlrp3tm1Bhk/JJ line was used with *Nlrp3* gene knockout: *NLRP3* KO, created at the Jackson Laboratory (USA) on the genetic background C57BL/6. Research on animals was carried out in accordance with the principles of humanity set forth in the European Community Directive (2010/63/EC) and with the permission of the Bioethical Commission of Professor V. F. Voino-Yasenetsky Krasnoyarsk State Medical University, Krasnoyarsk, Russia.

### 4.2. Behavioral Testing

#### 4.2.1. Open Field Test with Non-Social and Social Objects

The open field test is a widely used method to assess locomotor activity, exploration, and anxiety-like behavior in rodents, including mice. The open field test was performed as described previously in [72]. Briefly, we used a round open field arena (diameter 620 mm, wall height 320 mm) covered with gray polypropylene sheets. The central zone was a rounded field (diameter 300 mm). Each mouse was placed in the open field for 10 min. The test measured overall activity, time, and distance traveled in the central zone. After each test, the test chambers were treated with 70% alcohol. The number of animals used for the test was 4–6 individuals per experimental group.

After 10 min (habituation session) in the open field arena, we conducted the open field test with a non-social and a social object placed in the central zone to assess the fear of a new non-social (non-living) object and social preferences (social object). First, the non-social object was placed in the center of the arena. The mouse under study was placed in the open field, where the new non-social object (a cage) was located for the next 10 min.

Subsequently, the non-social object was replaced with another previously unfamiliar C56BL/6 male individual of the same age in a new cage (70 mm × 90 mm × 70 mm and bars spaced 5 mm apart). Testing was also carried out for 10 min. Time spent in the central zone, time spent with the objects, and approaches to the objects were analyzed in each session. At the end of each test, the open field and the non-social object were sprayed with 1% sodium hypochlorite followed by 70% ethanol and wiped with paper towels. The average time interval between sessions was 2–3 min. The entire testing process was recorded using the ANY MAZE animal video analysis system (Behavior Tracking Software Application Version 4.99m, Stoelting, Wood Dale, IL, USA).

#### 4.2.2. Fear Conditioning Test

The testing equipment consisted of an acrylic square chamber with an electrified grating floor for delivering an electrical signal (unconditioned stimulus, US) during white noise (conditioned stimulus, CS). The chamber was placed in a large soundproof box (Ugo Basil, Italy). The test was carried out according to the described protocol [73] for 3 days. The number of animals used for the test was 7–15 individuals per experimental group.

On the first day (conditioning day), conditions for memorization and learning were created, manifested by an increasing freezing time. Mice were placed in the conditioning chamber and were allowed to explore it for 120 s. Then, an auditory cue (white noise, 55 dB) was applied as a conditioned stimulus (CS) for 30 s, and a footshock of 0.3 mA was applied as an unconditioned stimulus (US) during the last 2 s of white noise. The presentation of CS–US stimuli was repeated three times in order to form a conditioned fear reflex. This combination of stimuli was presented at 120, 240, and 360 s after the beginning of the test. The analysis of the obtained results was carried out for each selected period of time (from 0 to 120 s, from 120 to 240 s, from 240 to 360 s, and from 360 to 480 s) to track the dynamics of acquiring the association of CS with US.

Contextual conditioning of fear (context day) was performed 24 h after the first test when freezing conditions were created in the same chamber for 300 s in the absence of any stimuli (no white noise (CS) and 0.3 mA footshock (US)).

On the third day (cued day), mice were placed in a chamber with a different wall color, floor structure (no grid), and 30 lux illumination, providing a new context. The test consisted of a 180-s period for mice to explore the new environment to assess non-specific contextual fear, followed by a 180-s conditioned stimulus (white noise (CS) without a footshock (US)) to assess conditioned fear. The percentage of freezing in each time period was considered the memory of fear since freezing behavior is often used as a quantitative characteristic in the fear conditioning test [31,74]. The entire testing process was recorded using the ANY MAZE animal video analysis system (Behavior Tracking Software, Stoelting, Wood Dale, IL, USA).

#### 4.2.3. Three-Chamber Sociability Test

The assessment of sociability and social preference in the experimental mice was conducted using a three-chamber box as previously described [8]. This apparatus comprises three interconnected polycarbonate chambers with small doorways allowing access into each section. The number of animals used for the test was 6–8 individuals per experimental group.

The test consisted of three phases. Initially, a 5-min habituation period was provided, during which the experimental mouse freely explored all chambers. In the subsequent sociability phase, the experimental mouse was presented with a choice between an enclosed novel mouse and an empty wire cup in the two separate chambers. The duration of time spent by the experimental mouse in the chamber containing the novel mouse compared to the chamber with the novel object was recorded over a 10-minute trial using video recording equipment and analyzed through ANY MAZE software.

Following the sociability phase, the preference for social novelty was assessed. This involved presenting the experimental mouse with a choice between the initially introduced mouse (stranger 1) and a second unfamiliar mouse (stranger 2) in the left and right chambers, respectively. The experimental mouse was given 10 min to explore the chambers, and the time spent in each chamber was recorded. Preference for social novelty was defined by the duration of time spent in the chamber with stranger 2 relative to the time spent in the chamber with stranger 1.

Between each experimental trial, the apparatus was meticulously cleaned with water and thoroughly dried. The purpose of these trials was to evaluate the sociability and preference for social novelty in the experimental mice under standardized conditions. Preference for the novel mouse was calculated as [(time spent exploring novel mouse)/(total time spent exploring novel mouse and novel object)] × 100%; preference for novel mouse 2 was calculated as [(time spent exploring novel mouse 2)/(total time spent exploring novel mouse 1 and novel mouse 2)] × 100%.

### 4.3. Slice Preparation and Electrophysiology

Mice (*n* = 3–6 per group) were euthanized by using carbon dioxide and decapitated, and then brains were removed and placed in ice-cold oxygenated (95% O_2_ and 5% CO_2_) Ringer’s solution (234 mM sucrose, 2.5 mM KCl, 1.25 mM NaH_2_PO_4_, 10 mM MgSO_4_, 0.5 mM CaCl_2_, 26 mM NaHCO_3_, 11 mM D-glucose) for 1 min. The brain hemispheres were isolated and sectioned using a vibratome (Thermo Fisher Scientific). Coronal mouse brain sections (*n* = 10–20 per group, 250 μm) were incubated for at least 1 h at room temperature (RT) in continuously oxygenated (95% O_2_ and 5% CO_2_) aCSF (125 mM NaCl, 2.5 mM KCl, 1.25 mM NaH_2_PO_4_, 1 mM MgCl_2_, 2 mM CaCl_2_, 26 mM NaHCO_3_, and 10 mM D-glucose). Registration of fEPSP was performed using a Cl–Ag electrode in a glass borosilicate pipette filled with aCSF. Stimulation of presynaptic fibers was performed using a platinum–iridium electrode. Potentials were recorded and stimulation was performed on the CA1 zone of the hippocampus. The CA1 zone was visually determined by shape with 10× magnification. Stimulation and registration were performed using the PatchMaster v2×90 software (HEKA, Lambrecht, Germany) and the pClamp10 amplifier (Molecular Devices, San Jose, CA, USA). The obtained data were analyzed using the Clampfit 10.5 program (Axon Instruments, Burlingame, CA, USA). Field excitatory postsynaptic potentials (fEPSPs) were evoked with pulses at a stimulation frequency of 0.33 Hz and at an intensity that could elicit 50% of the maximum fEPSP response. The rise time in the registration of fEPSPs on brain slices refers to the duration taken for the fEPSP to rise from a baseline to its peak amplitude. This measurement is important in assessing the kinetics and efficiency of synaptic transmission.

### 4.4. Cell Culture

Mice (*n* = 7–8 per group) were euthanized by using carbon dioxide and decapitated, and then brains were removed and placed in an ice-cold solution of 2% glucose in PBS (PanEco, Moscow, Russia), after which the hippocampus was isolated and dissected with a scalpel to a size of 1 mm^3^. The tissue pieces were transferred with a Pasteur pipette into a centrifuge tube containing a fresh solution of 2% glucose in PBS and left for 1 min. After sedimentation, the supernatant was removed. Then, 3 mL of NeuroCult NS-A Proliferation medium (StemCell, Cambridge, MA, USA) was added to the tube, and the tissue was mechanically dissociated with a sterile serological pipette until a homogeneous cell suspension was obtained, after which it was centrifuged at 150× *g* for 5 min. After centrifugation, the supernatant was removed and 3 mL of fresh NeuroCult NS-A Proliferation medium was added. The cell suspension was transferred into T-75 cm^2^ culture flasks with 25 mL of NeuroCult NS-A Proliferation medium. Cultivation was carried out in an incubator at 5% CO^2^ and 37 °C. After approximately 48 h, neurospheres were observed in the culture flasks. Passage and change of medium were carried out every 4–5 days. For the differentiation of neurospheres, at passage three, the medium was collected from the flasks, and the neurospheres were precipitated by centrifugation at 300 g for 5 min. The supernatant was discarded and the neurospheres were resuspended in DMEM culture medium (PanEco) supplemented with 20% FBS (HyClone, Logan, UT, USA), 0.58 mg/mL glutamine (PanEco), 100 U/mL penicillin, and 100 mg/mL streptomycin (PanEco). Cells were seeded into T-75 cm^2^ culture flasks, and after 5–7 days, the differentiation of neurospheres into a co-culture of astrocytes and neurons was observed. After 10 days, the differentiated cells were cultured into 24-well or 96-well culture plates for further study. We also confirmed the expression of the astrocyte marker GFAP and the neuronal marker NeuN in cell co-culture. The Appendix A contains high-quality images, providing visual evidence of the GFAP and NeuN (Appendix A).

### 4.5. Immunosenescence Study

Immunosenescence was studied using the Senescence Detection Kit (K320, BioVision, Waltham, MA, USA) according to the manufacturer’s protocol. The Senescence Detection Kit is designed to histochemically detect specific senescence markers in distinct pH in cultured cells and tissue sections. In vitro studies (cultured cells): The Senescence Detection Kit was employed for in vitro studies to assess senescence markers in cultured cells. This assay is specifically tailored to histochemically detect senescence markers within cultured cells by identifying distinct pH changes. Spectrophotometric measurements at 620 nm were used to quantify the intensity of the blue color appearance after staining (ZOE Fluorescent Cell Imager, Bio-Rad Laboratories, Hercules, CA, USA). The assay was conducted in adherence to the manufacturer’s protocol and was repeated a minimum of three times to ensure reliability and reproducibility. In vivo studies (sagittal sections): In the case of in vivo analysis using sagittal sections from mice (not fixed in PFA), we adapted the same principle of staining used for in vitro studies to histochemically detect senescence markers. The staining procedure was conducted following the same principle but involved the analysis of the sections using an Olympus BX45 microscope (Olympus Inc., Tokyo, Japan). Images were processed using ImageJ software, and a minimum of five randomly selected images per experimental group were evaluated to ensure representative analysis.

### 4.6. Immunostaining

Immunocytochemistry was performed according to the protocol of the antibody manufacturer. Primary antibodies used were IKK (ab178870, Abcam, Cambridge, UK), PKR (ab32506, Abcam), IR (ab137747, Abcam), IRS1 (ab66154, Abcam), IL-18 (sc6177-4, Santa Cruz Biotechnology, Dallas, TX, USA), HMGB (ab77302, Abcam), and NLRP3 (ab2307396, Abcam). Primary antibodies were used at a dilution of 1:300; incubation time was 18 h at 4 °C. Secondary antibodies used were Alexa Fluor 555 (ab150078, Abcam) and Alexa Fluor 488 (ab150117, Abcam). Secondary antibodies were used at a dilution of 1:500; incubation time was 2 h at 37 °C. Cell microscopy was performed using the ZOE fluorescent microscope (Bio-Rad, Hercules, CA, USA). The percentage of cells expressing antigen and fluorescence intensity were counted using the ImageJ program. This was determined by calculating the ratio of positive cells to the total cell count. There were 3 independent experiments with 2 wells per group in each experiment. At least 5 visual fields were analyzed in each well and the average values obtained per well were used for statistical analysis.

For immunohistochemistry, mice (*n* = 5–7 per group) were euthanized by using carbon dioxide and transcardially perfused with 4% paraformaldehyde (PFA) in 0.1 M PBS (Sigma, Ronkonkoma, NY, USA). Brains were removed, postfixed in 4% PFA at 4 °C overnight, and then immersed in 20% sucrose dissolved in PBS containing 0.01% sodium azide (Sigma) for 48 h at 4 °C. Sections 50 μm thick were made with a vibratome (Thermo Fisher Scientific, Waltham, MA, USA) in the sagittal direction using a stereotaxic atlas to localize the hippocampus. Free-floating brain sections were blocked in PBS containing 10% normal goat serum (Sigma), 2% BSA (Sigma), 1% Triton X-100 (Sigma), and 0.1% sodium azide (Sigma) for 1 h at RT. The following primary antibodies were used: GFAP (ab4674, Abcam), NeuN (ABN90, Merk, Whitehouse Station, NJ, USA), IRS1 (phosphor S312) (ab66154, Abcam), IL18 (sc6177, Santa Cruz Biotechnology, USA), NLRP3 (ab2307396, Abcam), IKKβ (ab178870, Abcam), and PKR (ab32506, Abcam). The day after incubation with the primary antibodies, the slices were washed in PBS and incubated with Alexa-conjugated secondary antibodies for 2 h at RT. After washing, the slices were mounted on glass slides, then the aqueous Fluoromount mounting medium (Sigma) was applied, and the slices were covered with coverslips. The images were acquired with a 60× objective on the Olympus FV 10i confocal fluorescence microscope (Olympus, Tokyo, Japan) and processed using Olympus FluoView (version 4.0a) and ImageJ (version 1.52a) software. Representative images of the hippocampus according to the Paxinos and Franklin stereotaxic atlas [75] were taken from at least 2 sections from 5–7 mice per group, and 5 fields of view were analyzed from each section. Using ImageJ, the area of immunopositive cells/field in µm^2^ was quantified for all the markers used. The area of selection in square pixels was calibrated to square units (µm^2^).

### 4.7. Insulin, Lactate, and IL-1β Quantitative Analyses

Insulin measurements were conducted using the insulin ELISA assay kit (Mouse Ultrasensitive Insulin ELISA 96 tests, 80-INSMSU-E01, ALPCO, Salem, NH, USA), following the manufacturer’s instructions. Hippocampus homogenates from 7 to 8 mice per group, prepared in phosphate-buffered saline (PBS), were utilized for the study. The assay kit has a sensitivity of 0.115 ng/mL. Protein concentration measurements were performed using the Bio-Rad protein assay kit, employing bovine serum albumin (BSA) as standards (Bio-Rad). The resulting protein concentration was reported in mg/mL. To calculate the concentration of insulin in each sample, the obtained values in ng/mL were divided by the corresponding protein concentration. Thus, the insulin evaluation in the homogenate was presented as ng/mg protein [74].

Lactate concentration in the hippocampus was analyzed using an enzymatic method followed by a colorimetric measurement. A commercially available kit (L-Lactate Assay Kit, ab65330, Abcam, UK) was utilized for this purpose [76]. The sensitivity of the kit is reported to be greater than 0.001 mM. Protein concentration measurements were determined using the Bio-Rad protein assay kit with bovine serum albumin as standards (Bio-Rad). The resulting protein concentrations were expressed in μg/μL. To calculate the concentration of lactate in each sample, the obtained values in nmol/μL were divided by the corresponding protein concentration. Consequently, the lactate concentration in the homogenate was presented as nmol/μg of protein.

IL-1β concentration was measured by ELISA according to the protocol provided in the IL-1β Mouse ELISA Kit (KMC0011). Hippocampus homogenates prepared in a buffer containing 5 M guanidine-HCl diluted in 50 mM Tris buffer, pH 8.0, and 1× PBS with 1× protease inhibitor were used for the analysis. The measurement was performed at a wavelength of 450 nm. The kit had a minimum detectable concentration of murine IL-1β of <7 pg/mL. Protein concentration measurements were conducted using the Bio-Rad protein assay kit with bovine serum albumin (BSA) as standards (Bio-Rad). The resulting protein concentration was reported in mg/mL. The IL-1β concentration in the homogenate was calculated by dividing the measured values in pg/mL by the protein concentration in each sample, yielding the IL-1β concentration in pg/mg protein.

### 4.8. Statistical Analysis

Statistical analysis of the obtained data was performed using the GraphPad Prism 10 program (GraphPad Software, La Jolla, CA, USA). The Kolmogorov–Smirnov test was used to assess the normality of the distribution. If the distribution was not normal, two groups were compared using the nonparametric Mann–Whitney U test. To assess the influence of two factors, two-way ANOVA was used. Subsequent pairwise comparison of groups was performed using Sidak’s post hoc test and Tukey’s post hoc test. Values were considered statistically significant at *p* ≤ 0.05. Values are presented as M ± SEM.

## 5. Conclusions

In this study, using genetic strategies, we described the significant function of the NLRP3 inflammasome in the control of aging-related chronic neuroinflammation and fear memory. Our data show that aging in mice is accompanied by the development of anxiety, impaired social activity, and impaired contextual memory associated with the hippocampus, but not in mice with *NLRP3* deletion. At the same time, adult *NLRP3* KO mice demonstrate reduced social activity and anxious behavior. Based on the data obtained in behavioral tests, it can be concluded that the basal (constitutional) level of inflammasome expression is necessary for normal social and pro-social behavior. At the same time, *NLRP3* KO protects against the signaling memory impairment during aging.

In addition, we showed that aging is accompanied by an increase in the number of senescent cells and an increase in the expression of inflammatory markers. Deletion of *NLRP3* does not lead to the development of the SASP phenotype during aging. Also, *NLRP3* KO leads to an increase in lactate levels, which may indicate that the absence of *NLRP3* probably increases glycolysis or lactate dehydrogenase activity or a combination of both mechanisms.

During aging in WT mice, phosphorylated metaflammasome proteinases IKK and PKR are expressed more. However, the expression of insulin receptor substrate 1 (IRS1phospho-S312) and insulin receptors (IR) does not alter with age in mice. The expression of the components of the metaflammasome is not increased by *NLRP3* gene deletion. Thus, in *NLRP3* KO mice, age-related metabolic changes can be stopped. In conclusion, our results indicate that *NLRP3* inhibition attenuates the effects of aging in the CNS. Deletion of *NLRP3* improves some behavioral and biochemical characteristics associated with aging, such as signal memory, glycolysis activity, neuroinflammation, and metabolic inflammation, but not anxious behavior. Mice lacking NLRP3 expression exhibited reduced interaction time with a social object compared to age-matched wild-type mice. The aging process in the absence of NLRP3 inflammasome activity did not affect sociability; however, it resulted in a diminished inclination towards social novelty. These results may be associated with reduced IL-18 signaling and the PKR/IKKβ/IRS1 pathway as well as the SASP phenotype. In *NLRP3* gene deletion conditions, PKR is down-regulated. These downstream responses could potentially influence IRS1 signaling and improve insulin sensitivity indirectly, impacting cellular metabolism and other functions related to insulin signaling. Therefore, it is likely that slowing aging through various NLRP3 inhibition mechanisms will lessen the corresponding cognitive decline during aging. Thus, the protective effects of genetic knockout of the NLRP3 inflammasome suggest a new therapeutic strategy for age-associated cognitive decline and for slowing down CNS aging.

### Limitations of the Study

The present study faced several experimental constraints and limitations, preventing the execution of specific analyses and assessments crucial to a comprehensive evaluation of the research objectives. In our study, staining for IKKbeta was not performed on neuronal cells, and PKR staining was not conducted on astroglial cells. It is advisable for forthcoming procedures to undertake marker detection on Iba-1+ cells. Data concerning IL-1β in the cell cultures and the expression of insulin receptors in vivo are not provided in our study. Nevertheless, in addressing this constraint, we assessed the expression of HMGB1 in cell culture experiments. However, we acknowledge the significance of conducting colocalization studies to further comprehend the interaction among different markers within the hippocampus.

## Figures and Tables

**Figure 1 ijms-24-16580-f001:**
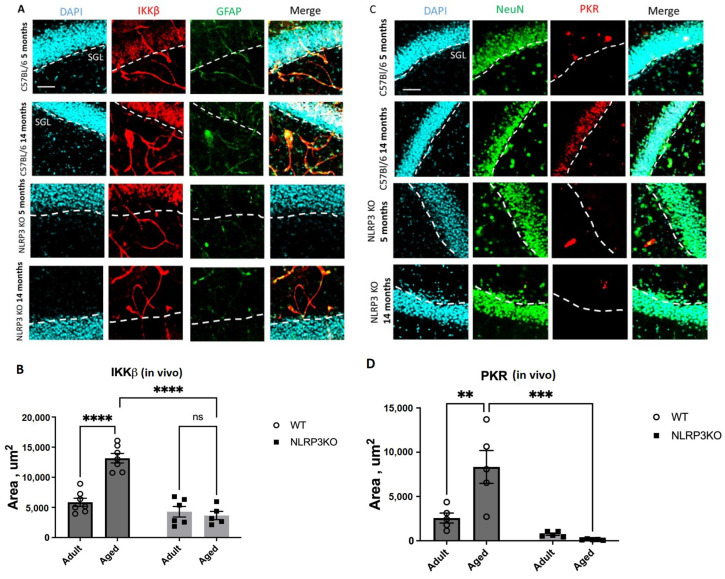
(**A**) Representative images of the hippocampus from adult (5 months) and aged (14 months) mice of different genotypes (WT and *NLRP3* KO) stained with IKKβ antibody (red) and GFAP antibody (green); cell nuclei were stained with DAPI (blue). WT—wild-type mice (C57Bl/6); *NLRP3* KO—knockout mice for the *NLRP3* gene. The scale bar is 100 µm, dashed lines—subgranular layer (SGL). (**B**) Area of IKKβ expression in the brain sections of adult (5 months) and aged (14 months) mice of different genotypes (WT and *NLRP3* KO), µm^2^. (**C**) Representative images of the hippocampus from adult (5 months) and aged (14 months) mice of different genotypes (WT and *NLRP3* KO) stained with PKR antibody (red) and NeuN antibody (green); cell nuclei were stained with DAPI (blue). WT—wild-type mice (C57Bl/6); *NLRP3* KO—knockout mice for the *NLRP3* gene. The scale bar is 100 µm, dashed lines—subgranular layer (SGL). (**D**) Area of PKR expression in the brain sections of adult (5 months) and aged (14 months) mice of different genotypes (WT and *NLRP3* KO), µm^2^. ns—not significant, **—*p* ≤ 0.01, ***—*p* ≤ 0.001, ****—*p* ≤ 0.0001, *n* = 5 to 7 mice per group. Values are presented as M ± SEM.

**Figure 2 ijms-24-16580-f002:**
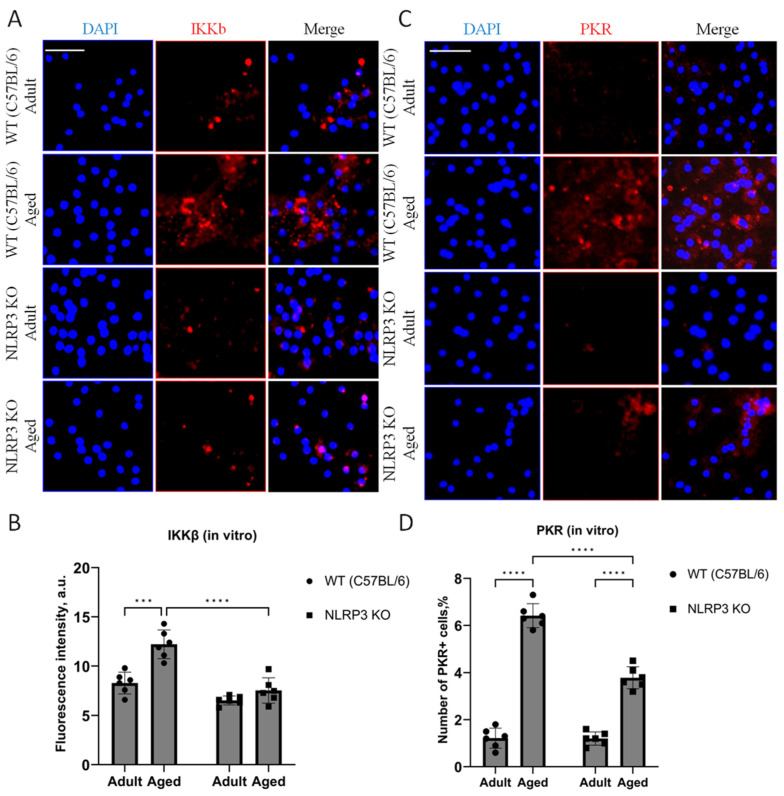
(**A**) Representative images of the co-cultures of astrocytes and neurons obtained from adult (5 months) and aged (14 months) mice of different genotypes: WT (wild-type mice (C57Bl/6)) and *NLRP3* KO (knockout mice for the *NLRP3* gene). Immunofluorescent staining with IKKβ antibody (red); cell nuclei were stained with DAPI (blue). The scale bar is 100 µm. (**B**) Fluorescence intensity of IKKβ in the co-cultures of astrocytes and neurons obtained from adult (5 months) and aged (14 months) mice of different genotypes (WT and *NLRP3* KO), a.u. ***—*p* ≤ 0.001, ****—*p* ≤ 0.0001. (**C**) Representative images of the co-cultures of astrocytes and neurons obtained from adult (5 months) and aged (14 months) mice of different genotypes: WT (wild-type mice (C57Bl/6)) and *NLRP3* KO (knockout mice for the *NLRP3* gene). Immunofluorescent staining with PKR antibody (red); cell nuclei were stained with DAPI (blue). The scale bar is 50 µm. (**D**) The number of PKR-immunopositive cells in the co-cultures of astrocytes and neurons obtained from adult (5 months) and aged (14 months) mice of different genotypes (WT and *NLRP3* KO), %. ****—*p* ≤ 0.0001. Values are presented as M ± SEM.

**Figure 3 ijms-24-16580-f003:**
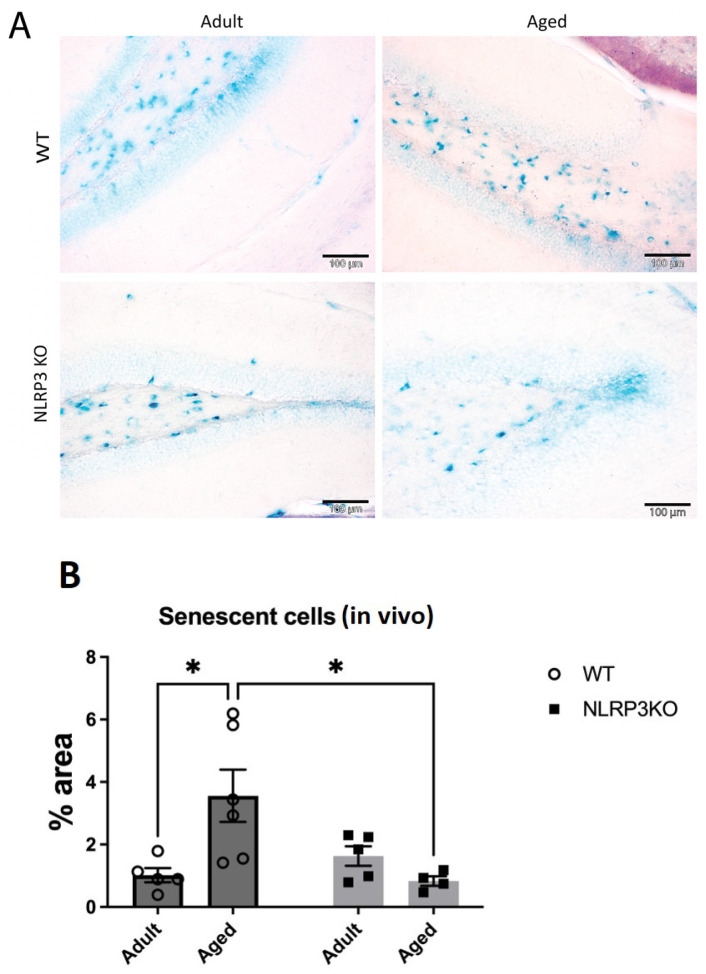
(**A**) Representative images of the hippocampus from adult (5 months) and aged (14 months) mice of different genotypes (WT and *NLRP3* KO) stained with SA-β-gal. Blue staining identifies senescent cells. WT—wild-type mice (C57Bl/6); *NLRP3* KO—knockout mice for the *NLRP3* gene. The scale bar is 100 µm. (**B**) Quantitative analysis of SA-β-gal-positive cells presented as an occupied area (%) on hippocampal slices. WT—wild-type mice (C57Bl/6); *NLRP3* KO—knockout mice for the *NLRP3* gene; Adult—mice at the age of 5 months; Aged—mice at the age of 14 months. *—*p* ≤ 0.05. *n* = 5–6 mice per group. Values are presented as M ± SEM.

**Figure 4 ijms-24-16580-f004:**
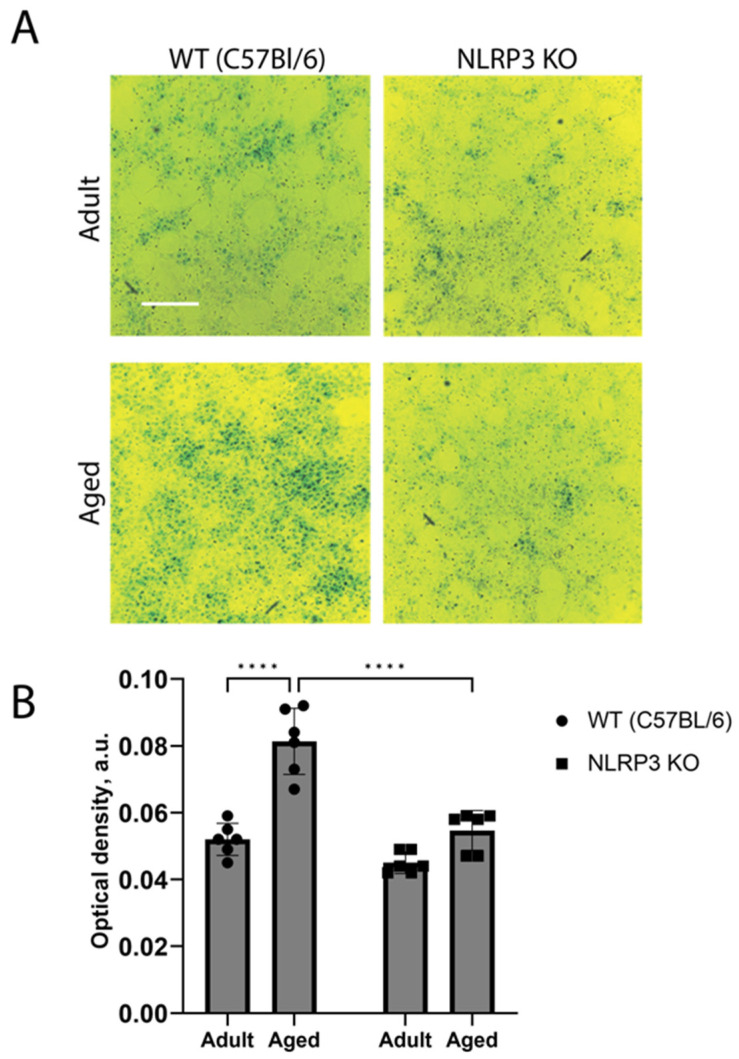
(**A**) Representative images of the co-cultures of astrocytes and neurons obtained from adult (5 months) and aged (14 months) mice of different genotypes: WT (wild-type mice (C57Bl/6)) and *NLRP3* KO (knockout mice for the *NLRP3* gene). Staining for SA-β-gal (green). The scale bar is 100 µm. (**B**) Quantitative analysis of SA-β-gal-positive cells presented as optical density (a.u.) in the co-cultures of astrocytes and neurons obtained from adult (5 months) and aged (14 months) mice of different genotypes. WT—wild-type mice (C57Bl/6); *NLRP3* KO—knockout mice for the *NLRP3* gene. ****—*p* ≤ 0.0001. Values are presented as M ± SEM.

**Figure 5 ijms-24-16580-f005:**
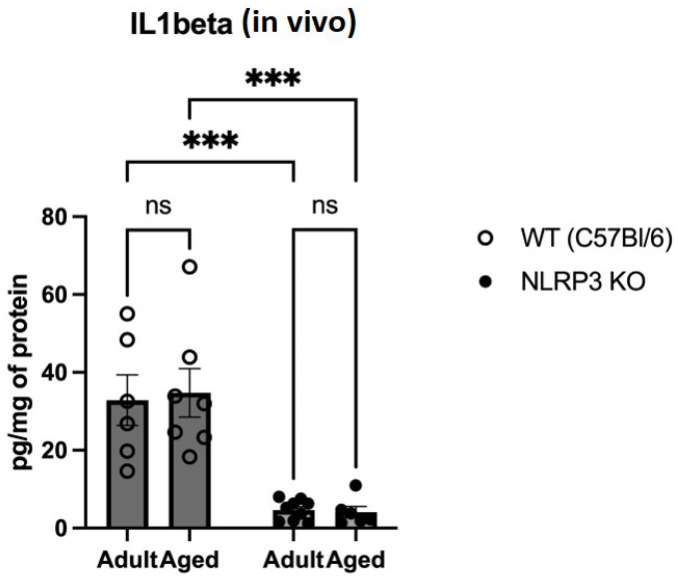
Determination of IL1 beta (IL-1β) levels in the hippocampal homogenates obtained from adult (5 months) and aged (14 months) mice of different genotypes: WT (wild-type mice (C57Bl/6)) and *NLRP3* KO (knockout mice for the *NLRP3* gene), pg/mg of protein. ns—not significant, ***—*p* ≤ 0.001. *n* = 6 to 9 mice per group. Values are presented as M ± SEM. IL-1β concentration was measured using ELISA according to the protocol provided in the IL-1β Mouse ELISA Kit.

**Figure 6 ijms-24-16580-f006:**
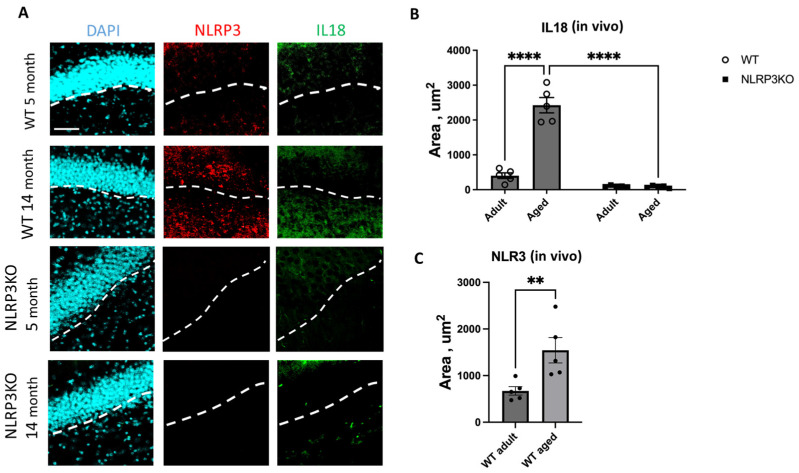
(**A**) Representative images of the hippocampus from adult (5 months) and aged (14 months) mice of different genotypes (WT and *NLRP3* KO) stained with NLRP3 antibody (red) and IL18 antibody (green); cell nuclei were stained with DAPI (blue). WT—wild-type mice (C57Bl/6); *NLRP3* KO—knockout mice for the *NLRP3* gene. The scale bar is 100 µm, dashed lines—subgranular layer (SGL). (**B**) Area of IL18 expression in the brain sections of adult (5 months) and aged (14 months) mice of different genotypes (WT and *NLRP3* KO), µm^2^. (**C**) Area of NLRP3 expression in the brain sections of adult (5 months) and aged (14 months) mice of different genotypes (WT and *NLRP3* KO), µm^2^. **—*p* ≤ 0.01, ****—*p* ≤ 0.0001, *n* = 5 to 7 mice per group. Values are presented as M ± SEM.

**Figure 7 ijms-24-16580-f007:**
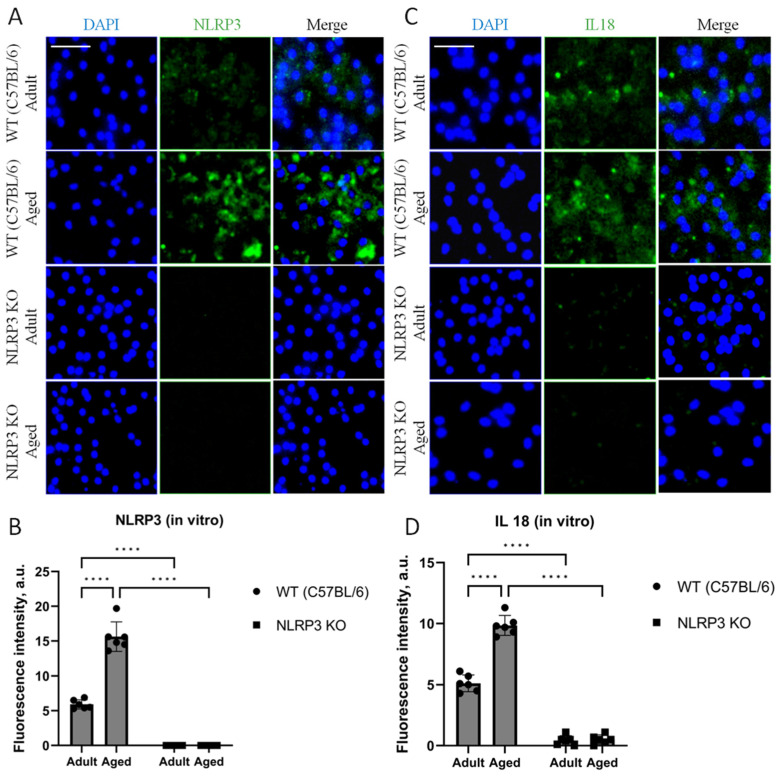
(**A**) Representative images of the co-cultures of astrocytes and neurons obtained from adult (5 months) and aged (14 months) mice of different genotypes: WT (wild-type mice (C57Bl/6)) and *NLRP3* KO (knockout mice for the *NLRP3* gene). Staining for NLRP3 antibody (green); cell nuclei were stained with DAPI (blue). The scale bar is 100 µm. (**B**) Fluorescence intensity of NLRP3 in the co-cultures of astrocytes and neurons obtained from adult (5 months) and aged (14 months) mice of different genotypes (WT and *NLRP3* KO), a.u. ****—*p* ≤ 0.0001. (**C**) Representative images of the co-cultures of astrocytes and neurons obtained from adult (5 months) and aged (14 months) mice of different genotypes: WT (wild-type mice (C57Bl/6)) and *NLRP3* KO (knockout mice for the *NLRP3* gene). Staining for IL-18 antibody (green); cell nuclei were stained with DAPI (blue). The scale bar is 50 µm. (**D**) Fluorescence intensity of IL-18 in the co-cultures of astrocytes and neurons obtained from adult (5 months) and aged (14 months) mice of different genotypes (WT and *NLRP3* KO), a.u. ****—*p* ≤ 0.0001. Values are presented as M ± SEM.

**Figure 8 ijms-24-16580-f008:**
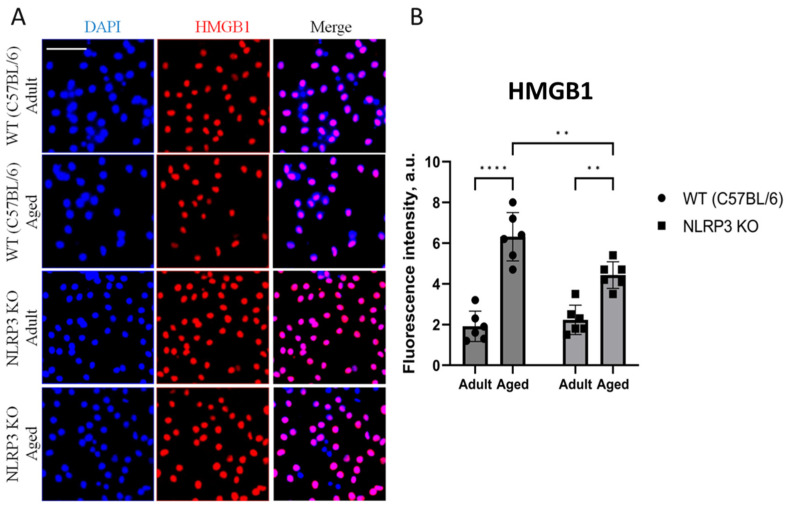
(**A**) Representative images of the co-cultures of astrocytes and neurons obtained from adult (5 months) and aged (14 months) mice of different genotypes: WT (wild-type mice (C57Bl/6)) and *NLRP3* KO (knockout mice for the *NLRP3* gene). Staining for HMGB1 antibody (red); cell nuclei were stained with DAPI (blue). The scale bar is 100 µm. (**B**) Fluorescence intensity of HMGB1 in the co-cultures of astrocytes and neurons obtained from adult (5 months) and aged (14 months) mice of different genotypes (WT and *NLRP3* KO), a.u. **—*p* ≤ 0.01, ****—*p* ≤ 0.0001. Values are presented as M ± SEM.

**Figure 9 ijms-24-16580-f009:**
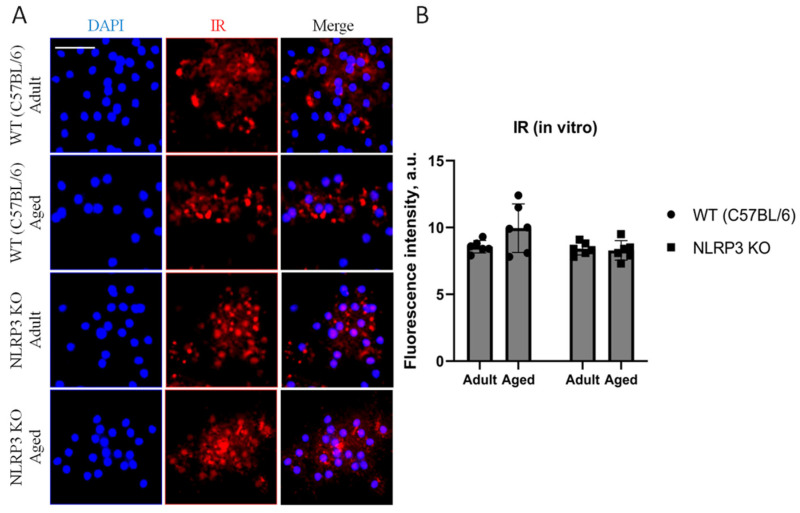
(**A**) Representative images of the co-cultures of astrocytes and neurons obtained from adult (5 months) and aged (14 months) mice of different genotypes: WT (wild-type mice (C57Bl/6)) and *NLRP3* KO (knockout mice for the *NLRP3* gene). Staining for IR antibody (red); cell nuclei were stained with DAPI (blue). The scale bar is 50 µm. (**B**) Fluorescence intensity of IR in the co-cultures of astrocytes and neurons obtained from adult (5 months) and aged (14 months) mice of different genotypes (WT and *NLRP3* KO), a.u. Values are presented as M ± SEM.

**Figure 10 ijms-24-16580-f010:**
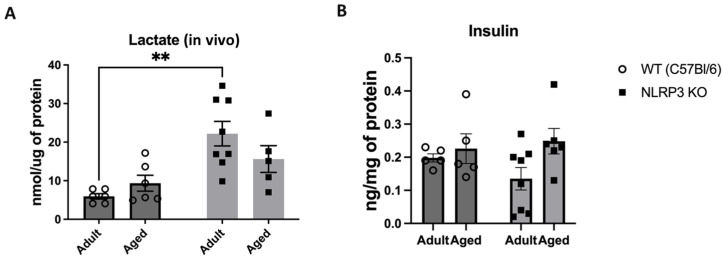
(**A**) Determination of lactate levels in the hippocampal homogenates obtained from adult (5 months) and aged (14 months) mice of different genotypes: WT (wild-type mice (C57Bl/6)) and *NLRP3* KO (knockout mice for the *NLRP3* gene), nmol/µg of protein. (**B**) Determination of insulin levels in the hippocampal homogenates obtained from adult (5 months) and aged (14 months) mice of different genotypes: WT (wild-type mice (C57Bl/6)) and *NLRP3* KO (knockout mice for the *NLRP3* gene), ng/mg of protein. **—*p* ≤ 0.01. *n* = 5 to 8 mice per group. Values are presented as M ± SEM. Lactate concentration in the hippocampus was analyzed using an enzymatic method followed by a colorimetric measurement, and insulin measurements were conducted using the insulin ELISA assay kit.

**Figure 11 ijms-24-16580-f011:**
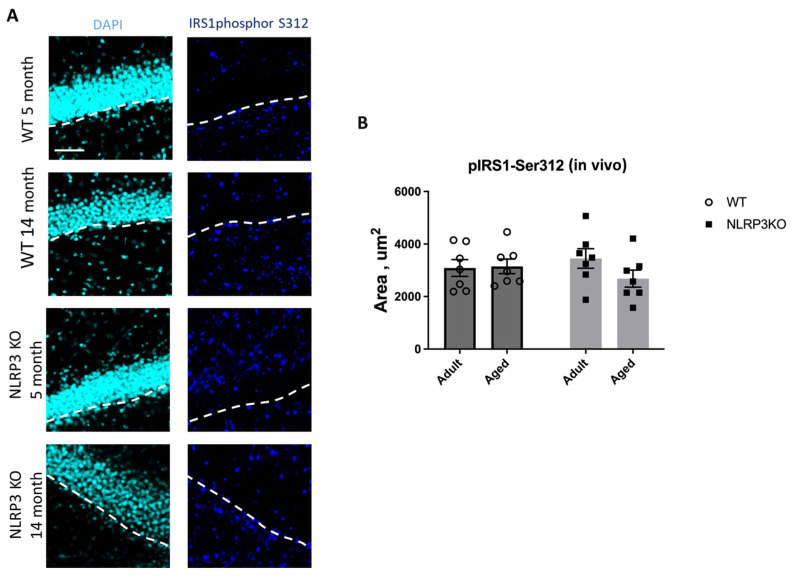
(**A**) Representative images of the hippocampus from adult (5 months) and aged (14 months) mice of different genotypes (WT and *NLRP3* KO) stained with IRS1phospho-S312 antibody (cyan); cell nuclei were stained with DAPI (blue). The subgranular zone (SGZ) is indicated as a dotted white line. The scale bar is 100 µm, dashed lines—subgranular layer (SGL). (**B**) Quantitative analysis of the IRS1phospho-S312 expression area on hippocampal slices, µm^2^. WT—wild-type mice (C57Bl/6); *NLRP3* KO—knockout mice for the *NLRP3* gene; Adult—mice at the age of 5 months; Aged—mice at the age of 14 months. *n* = 7 mice per group. Values are presented as M ± SEM.

**Figure 12 ijms-24-16580-f012:**
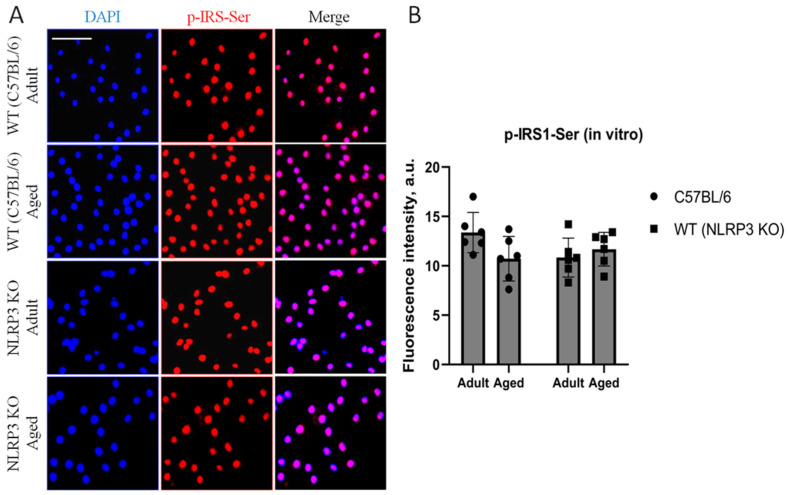
(**A**) Representative images of the co-cultures of astrocytes and neurons obtained from adult (5 months) and aged (14 months) mice of different genotypes: WT (wild-type mice (C57Bl/6)) and *NLRP3* KO (knockout mice for the *NLRP3* gene). Staining for pIRS1-Ser, an insulin receptor substrate (red); cell nuclei were stained with DAPI (blue). The scale bar is 50 µm. (**B**) Fluorescence intensity of pIRS1-Ser in the co-cultures of astrocytes and neurons obtained from adult (5 months) and aged (14 months) mice of different genotypes (WT and *NLRP3* KO), a.u. Values are presented as M ± SEM.

**Figure 13 ijms-24-16580-f013:**
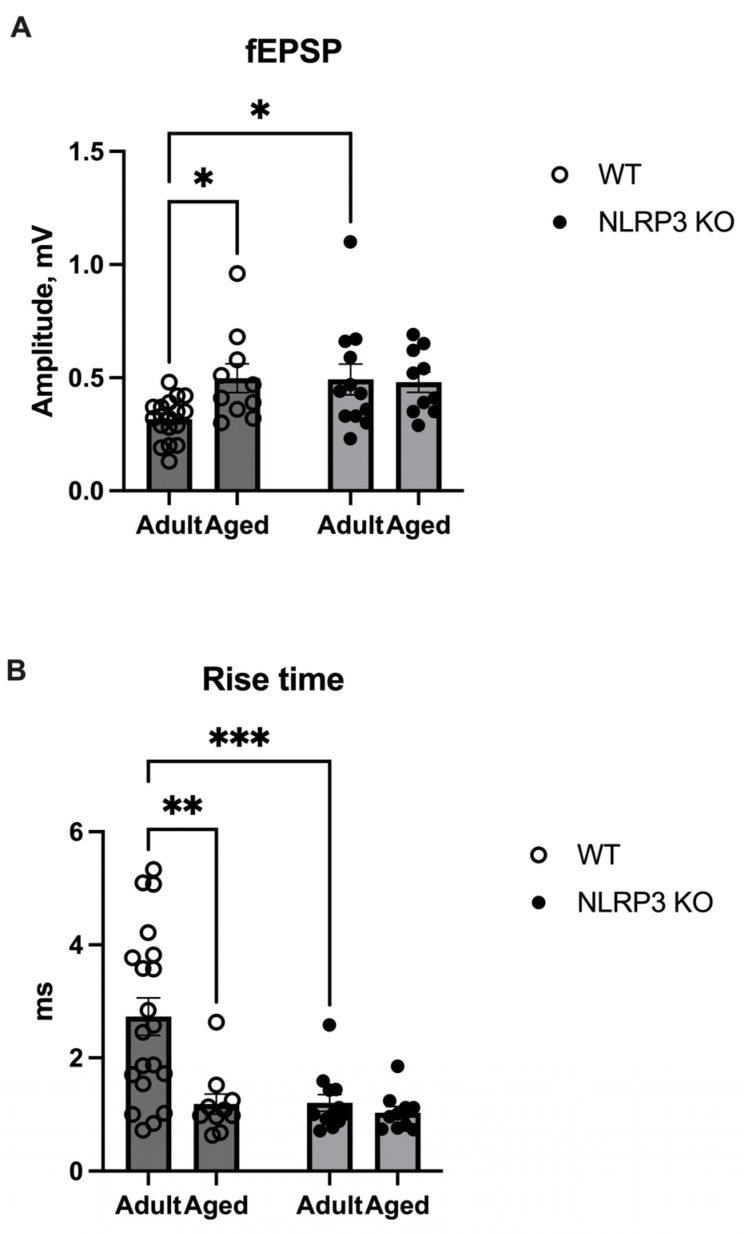
(**A**) Amplitude of field excitatory postsynaptic potentials (fEPSPs) in the CA1 zone of the hippocampus in adult (5 months) and aged (14 months) mice of different genotypes: WT (wild-type mice (C57Bl/6)) and *NLRP3* KO (knockout mice for the *NLRP3* gene), mV. (**B**) Rise time of the fEPSP amplitudes in the CA1 zone of the hippocampus in adult (5 months) and aged (14 months) mice of different genotypes: WT (wild-type mice (C57Bl/6)) and *NLRP3* KO (knockout mice for the *NLRP3* gene), ms. *—*p* ≤ 0.05, **—*p* ≤ 0.01, ***—*p* ≤ 0.001. *n* = 3 to 6 mice per group. Values are presented as M ± SEM.

**Figure 14 ijms-24-16580-f014:**
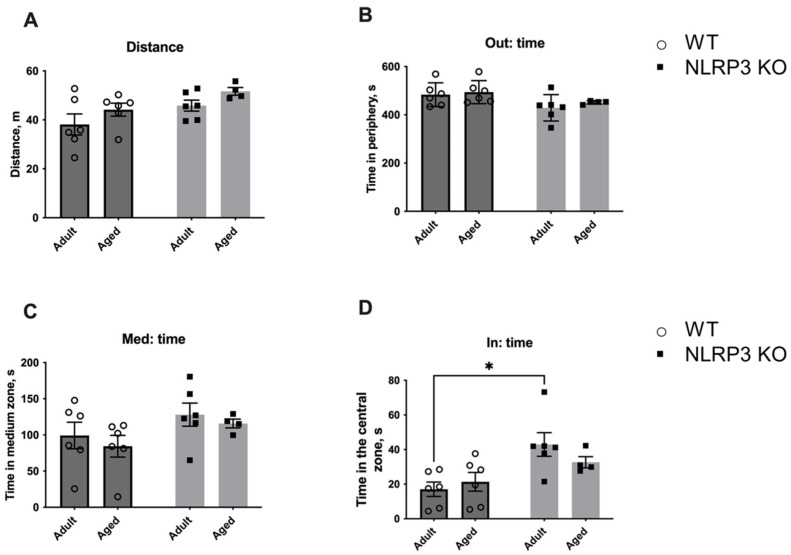
First session of the open field test. (**A**) Traveled distance, m. (**B**) Time spent in the periphery, s. (**C**) Time spent in the medium zone, s. (**D**) Time spent in the central zone, s. WT—wild-type mice (C57Bl/6); *NLRP3* KO—knockout mice for the *NLRP3* gene; Adult—mice at the age of 5 months; Aged—mice at the age of 14 months. *—*p* ≤ 0.05. *n* = 5 to 6 mice per group. Values are presented as M ± SEM.

**Figure 15 ijms-24-16580-f015:**
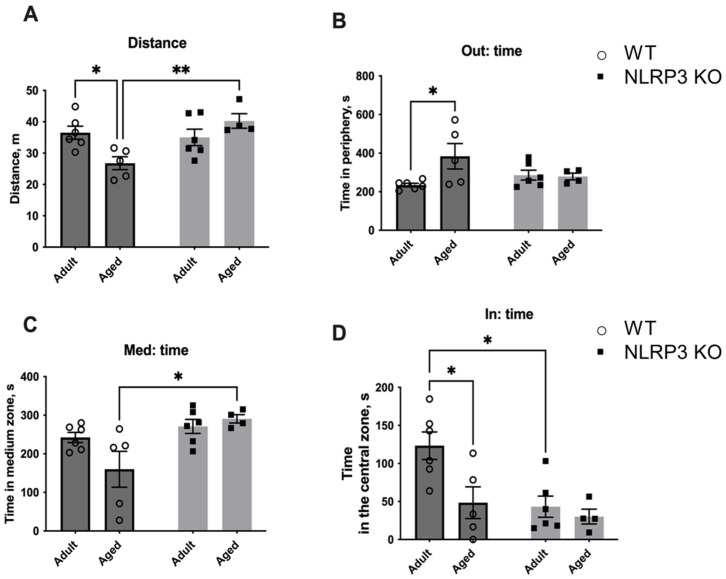
Second session of the open field test (with a non-social object). (**A**) Traveled distance, m. (**B**) Time spent in the periphery, s. (**C**) Time spent in the medium zone, s. (**D**) Time spent in the central zone, s. WT—wild-type mice (C57Bl/6); *NLRP3* KO—knockout mice for the *NLRP3* gene; Adult—mice at the age of 5 months; Aged—mice at the age of 14 months. *—*p* ≤ 0.05, **—*p* ≤ 0.01. *n* = 5 to 6 mice per group. Values are presented as M ± SEM.

**Figure 16 ijms-24-16580-f016:**
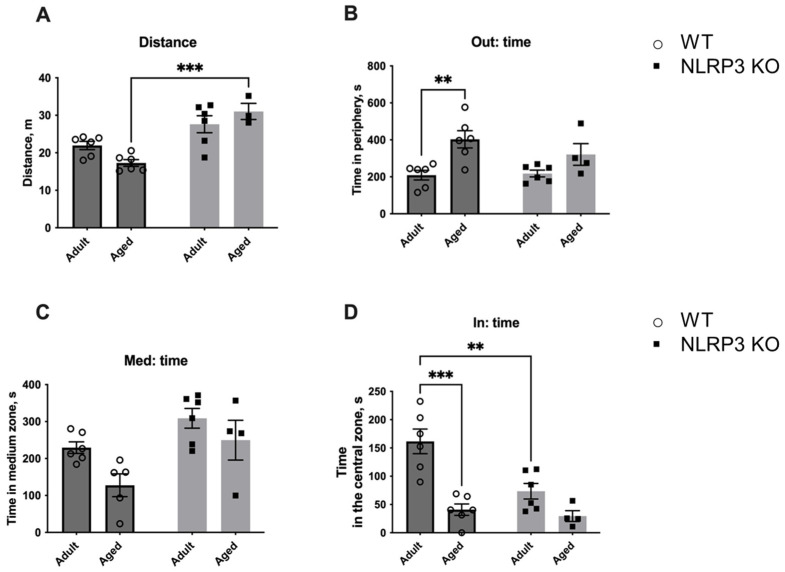
Third session of the open field test (with a social object). (**A**) Traveled distance, m. (**B**) Time spent in the periphery, s. (**C**) Time spent in the medium zone, s. (**D**) Time spent in the central zone, s. WT—wild-type mice (C57Bl/6); *NLRP3* KO—knockout mice for the *NLRP3* gene; Adult—mice at the age of 5 months; Aged—mice at the age of 14 months. **—*p* ≤ 0.01, ***—*p*≤ 0.001. *n* = 4 to 6 mice per group. Values are presented as M ± SEM.

**Figure 17 ijms-24-16580-f017:**
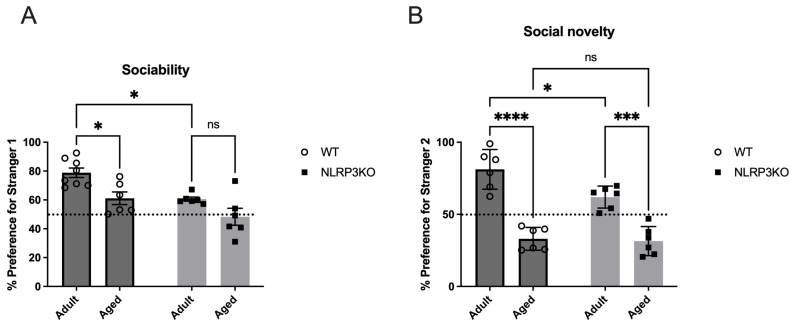
Social behavior in the three-chamber sociability test in *NLRP3* KO and WT mice. Duration measures were taken for (**A**) sociability, i.e., preference for a stranger mouse vs. an empty chamber, and (**B**) social novelty, i.e., preference for a novel stranger (stranger 2) vs. the first unfamiliar mouse (stranger 1). WT—wild-type mice (C57Bl/6); *NLRP3* KO—knockout mice for the *NLRP3* gene; Adult—mice at the age of 5 months; Aged—mice at the age of 14 months. ns—not significant, *—*p* ≤ 0.05, ***—*p* ≤ 0.001, ****—*p* ≤ 0.0001, *n* = 6 to 8 mice per group. Values are presented as M ± SEM.

**Figure 18 ijms-24-16580-f018:**
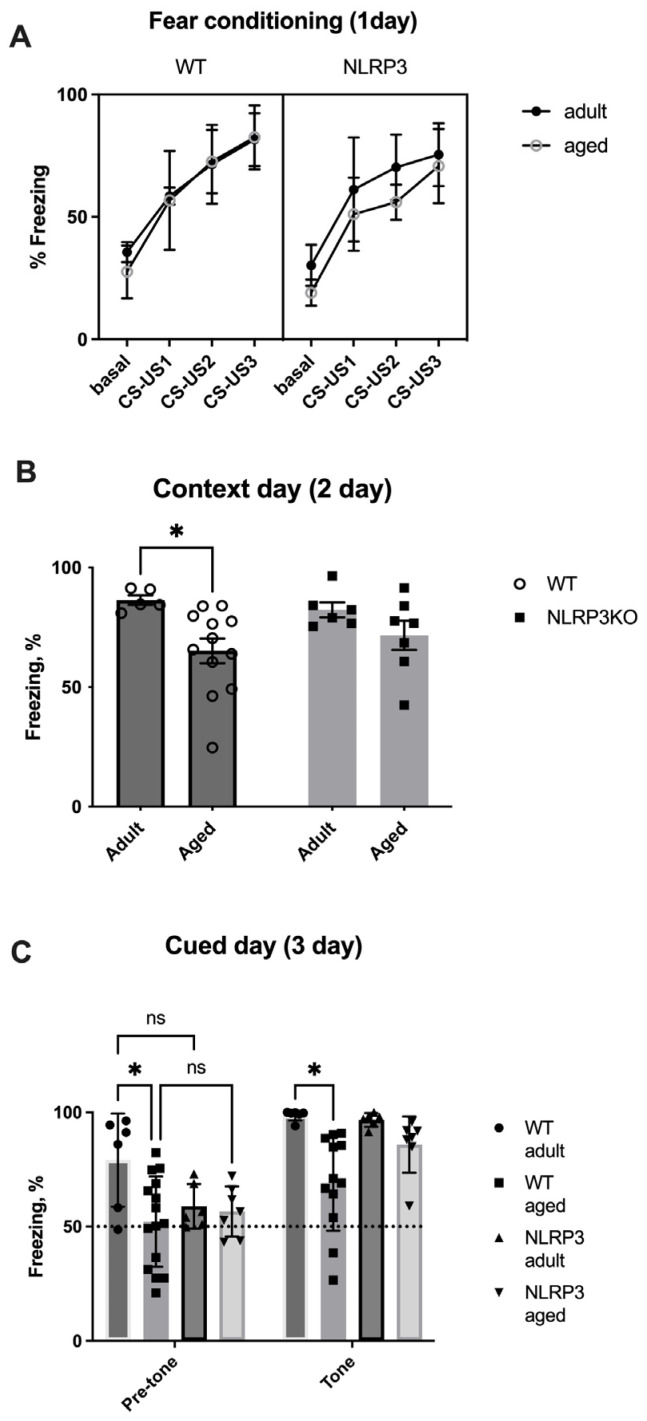
(**A**) Percentage of freezing time on the first day of the fear conditioning test. Basal—before white noise; CS-US1—the first pair of conditioned and unconditioned stimuli; CS-US2—the second pair of conditioned and unconditioned stimuli; CS-US3—the third pair of conditioned and unconditioned stimuli. (**B**) Number of freezing episodes on the first day of the fear conditioning test. (**C**) Percentage of freezing time on the third day of the fear conditioning test. WT—wild-type mice (C57Bl/6); *NLRP3* KO—knockout mice for the *NLRP3* gene; Adult—mice at the age of 5 months; Aged—mice at the age of 14 months. ns—not significant, *—*p* ≤ 0.05. *n* = 7 to 15 mice per group. Values are presented as M ± SEM.

## Data Availability

All data generated or analyzed during this study are included in this article. Further inquiries can be directed to the senior corresponding author.

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
