# Peer review of "Impact of NLRP3 Depletion on Aging-Related Metaflammation, Cognitive Function, and Social Behavior in Mice"

_ijms, 2023, doi:10.3390/ijms242316580_

Round 1

Reviewer 1 Report

Comments and Suggestions for Authors

In this manuscript, the authors conducted a sophisticated investigation into the role of the NLRP3 inflammasome in brain aging and behaviors. Remarkably, NLRP3 knockout demonstrated a protective effect against aging, suggesting a promising avenue for CNS aging treatment. The overall concept is captivating, and the findings bear substantial potential significance. Therefore, with the following minor revisions, this paper can be accepted:

  1. The study associates the observed changes in fEPSP parameters with altered neurotransmitter release from the presynapse. Could there be other factors, such as postsynaptic changes in receptor sensitivity or synaptic vesicle dynamics, that contribute to these alterations in synaptic transmission? How have these possibilities been considered?

  1. Given that synaptic plasticity is a complex process involving multiple forms and mechanisms, does the study consider the potential selectivity of lactate's effects on specific types of plasticity, such as long-term potentiation (LTP) or long-term depression (LTD)? Could lactate influence different types of plasticity differently?

  1. While the findings are well-presented, it would be valuable to discuss a little bit about the practical applications for therapeutic and preventive approaches in aging-related diseases, especially considering the NLRP3 inflammasome's role in immune defense. Addressing challenges and potential disparities between in vivo and in vitro data could enhance the findings' robustness and relevance.

Author Response

Dear Reviewer,

We sincerely appreciate your high evaluation of our manuscript on the role of the NLRP3 inflammasome in brain aging and behaviors. Your comments on electrophysiological issues and observations have been invaluable in refining our study and strengthens of discussion session.

We deeply value your constructive feedback and assure you that we will diligently address these points in the revised manuscript. Your guidance on two important points enhanced the quality and depth of our research. We hope that the forthcoming version will satisfactorily address these concerns.

Below are answers to all your comments and questions (attached).

Thank you once again for your time and thoughtful consideration of our work.

Best regards,

Yulia Komleva

Reviewer 2 Report

Comments and Suggestions for Authors

In this paper the authors aimed to study the underlying mechanisms of cognitive impairment and neuroinflammation related to aging. They addressed the possible involvement of NLRP3 in some social and cognitive behavioral aspects (data which may provide real novelty) and neuroinflammation associated with aging (already quite well stablished). Also, they tried to stablish a connection between NLRP3 and insulin signaling pathways, which they couldn´t find.

To achieve their goals they performed quite a number of experiments.

However, some points need to be reviewed.

Major points:

-        Methods:

o   In the “statistical analyses” section authors state: “Values are presented as M ± SEM”. If this is the case, the statistics of quite a few results showing “statistically significant differences” should be reviewed (at least fig3B, fig5, fig12D, 13B, C y D, fig14B, fig15B y C). Either, there is a mistake in the SEM or in those graphs some data are not statistically significant differences.

o   Number of animals used for every experiment with mice and number of repetitions done for co-culture experiments should be mentioned. Also, number of animals used for behavioral tests have to be clear.

o   Every graph should be provided as a dot plot graph to show data distribution by plotting dots for each observation or animal tested.

o   In the figure legend of graphs, how values are represented (M ± SEM?) should be indicated.

o   In the paragraph about “immunostaining”,  

§  Line 793: “A relative number of the cells expressing the antigen and fluorescence intensity were calculated using the ImageJ software.”

What does “a relative number” mean? Please explain better how the quantification was done. It should be at least from 3 independent experiments. How many images, from how many wells per group were analyzed?

§  How the area of expression of IKKb, PKR and IRS1 was determined and quantified should be clearly explained. How many images from each section were used for the quantification should be mentioned. How was the data finally analyzed?

o   In the paragraph about “Immunosenescence Study” it is a requirement to specify:

§  the reason why authors used different methods for in vivo and in vitro experiments.

§  number of cultures performed (at least form 3 independent experiments), number of wells per group, number of images analyzed.

§  and number of animals, how many slices and pictures were taken for these analyzes.

-        Results:

o   Figs 2, 6, 7 and 10: images showing immunostaining data should be provided as in Fig1 and 9.

o   Fig 1: Please show the images counterstained with NeuN both for IKKb and PKR.

§  Authors should mention within the text what cells showed double staining with GFAP and/or NeuN and discuss these results.

o   Fig 2: please, show counterstaining with GFAP and NeuN as in fig. 1 and provide higher magnification images. Discuss colocalizations.

o   Fig 3B. p= 0.0269 is not correctly indicated.

o   Fig 5. IL1b doesn’t show significant differences between adult and aged mice. However, within the text one can read: “In multiple comparisons, it was found that in the 281 hippocampal homogenates of aged WT animals IL-1β was significantly higher (39.05 ± 282 8.12 pg/mg) compared to the group of adult WT mice (28.45 ± 7.11 pg/mg)”. What is correct?

o   Fig 5 and fig 8. Please, indicate the method used for the IL1b, Lactate and Insulin analyzes withing the figure legend.

o   In lines 278-279 authors say: “we studied changes in the expression of IL-1β, as well as the expression of NLRP3…” They analyzed the expression of IL-1β in hippocampal homogenates and NLRP3 and IL-18 in co-cultures of neurons and astrocytes.

Moreover, in lines 293-294: “In our cell culture experiments, we analyzed the expression of NLRP3 and IL-18. The results obtained in the cell culture showed similar patterns of expression as observed in in vivo data.”

To be coherent, the authors should, at least, provide the immunostaining for NLRP3 and IL-18 in hippocampal slices. It would be very good to have also IL-1β expression in cultures.

o   Furthermore, IR immunostaining should also be shown in hippocampal slices.

o   Counterstaining with GFAP and NeuN for NLRP3, IL18 and IR should also be shown in the new figure as in fig. 1 and colocalizations discussed.

o   Graphs show two different patterns: 1. As in fig1; 2. As in fig 2. They all should follow the same pattern for comparisons among adult and aged mice and WT and NLRP3 deficient mice.

o   Please, be consistent in indicating of the p-value in every figure.

-        Discussion:

o   A main aim of this work was “to investigate the disruption of insulin signaling in the brains of wild-type (WT) and NLRP3 knockout (KO) mice of different ages”.

The conclusion given in lines 604-606: “There was no significant change in the expression of insulin receptors, which suggests that during physiological aging metabolic inflammation is observed earlier than insulin signaling impairment”. This conclusion doesn´t meet the authors first expectations and needs to be further discussed.

o   In line 880 can be read: “These results may be associated with reduced IL-18 signaling and the PKR/IKKβ/IRS1 pathway as well as the SASP phenotype.” Further discussion between the relation of PKR/IKKβ and IRS1 pathway would be desirable.

o   In lines 872-873 can be read: “while there may not yet be insulin signaling violations during physiological aging without indications of neurodegeneration and reactive astrogliosis, manifestations of metabolic inflammation may already be present.”

Comments on this are missing in the results section and they have to be added.

o   Lines 877-878 say: ”Deletion of NLRP3 improves behavioral and biochemical characteristics associated with aging, such as signal memory, anxiety, social function,??? glycolysis activity, neuroinflammation, and metabolic inflammation”. This is not coherent with data shown in the “results” section: line 412: “NLRP3 Deletion Leads to Anxious Behavior and Impaired Social Activity in Adult Animals.” Please, correct.

-        Title: the title doesn’t give clear information of what the paper is about, specially, taking into account that the behavioral data needs to be clarified.

-        The summary: once the statistical questions regarding behavioral tests have been answered, the summary may need to be revised.

Minor points:

-        Please, revise the introduction of abbreviations the first time they are used within the manuscript.

-        In graphs showing data of “area of marker expression in brain sections”, give correct units (um2??).

-        In methods: explain how “rise time” was analyzed in behavioral tests.

-        Line 407: “Significant effect of the time factor (F(3,87) = 112.9, p < 487 0.0001) and object matching (F(29,87) = 3.844, p < 0.0001) were revealed (Figure 15A).” In which group of mice? Please, show in Figure 15A (it seems missing).

-        Line 667: “we confirmed increased fear in NLRP3-/- mice with preserved high motor activity”. How had motor activity been tested?

-        “Typo” in line 811: the Olympus FluoView (Ver.4.0a) software and ImageJ software.

-        The information on “NLRP3 Deletion Leading to Anxious Behavior and Impaired Social Activity in Adult and aged Animals” is missing in the summary and should be added.

Comments on the Quality of English Language

English is understandable. It only needs minor editing.

Author Response

Dear Reviewer,

We extend our gratitude for your thorough assessment of our manuscript and the invaluable feedback provided. Your comments have significantly contributed to the refinement and enrichment of our study.

We have taken your suggestions into careful consideration and have made extensive revisions, incorporating additional information and newly added data on behavioral analyses, IHC, and other pertinent sections. These revisions aim to strengthen the depth and quality of the manuscript.

Your thoughtful guidance has been instrumental in shaping the enhancements made to the revised version of the paper. We are optimistic that these changes will address the concerns raised and improve the overall quality and comprehensiveness of the study.

Your commitment to ensuring the scientific rigor and accuracy of our work is deeply appreciated. We believe that your input has been pivotal in advancing the scholarly value of our research.

Once again, thank you for dedicating your time and expertise to meticulously review our manuscript. We remain committed to addressing your feedback and ensuring that the revised version meets the expected standards.

We eagerly await your evaluation of the updated manuscript and hope that the changes made align with your expectations. Your continued guidance is immensely valuable to us.

Below are answers to all your comments and questions (attached).

Thank you for your valuable feedback and guidance in improving the quality of our manuscript.

Best wishes,

Yulia Komleva

Round 2

Reviewer 2 Report

Comments and Suggestions for Authors

The authors satisfactorily answered almost every experiment, point and question which were requested. The paper has been significantly improved its quality, especially regarding the methodology used and the discussion provided for their results.

Still, graphs quantifying immunostaining of cell culture show 6 dots per group when they state in the section of  “methods” they performed:

Line 1048-49: 3 independent experiments with three wells per group in each experiment. At least 5 visual fields were analyzed in each well.

In my opinion they should say how they obtained the 6 dots per group.

Author Response

Summary                   

The authors thank the reviewer for taking the time to review the manuscript and for appreciating its revised version. Please find the detailed response below and the corresponding revisions in the re-submitted files.

Comment 1: The authors satisfactorily answered almost every experiment, point and question which were requested. The paper has been significantly improved its quality, especially regarding the methodology used and the discussion provided for their results.

Still, graphs quantifying immunostaining of cell culture show 6 dots per group when they state in the section of  “methods” they performed:

Line 1048-49: 3 independent experiments with three wells per group in each experiment. At least 5 visual fields were analyzed in each well.

In my opinion they should say how they obtained the 6 dots per group.

Response: Thank you for pointing this out. The authors agree that there is inconsistency in the graphs representing the results and in the description of the methods. We have revised the explanation of our quantification method in the section “4.6. Immunostaining” on p. 29-30 (Lines 1048-51, blue text). There were two wells used per each experimental group in three independent experiments. The six dots represent average values obtained from 5 visual fields per well, hence: 2 (wells) * 3 (experiments) = 6 dots (average values).